# Evolution of drift robustness in small populations

Thomas LaBar[1,2,3] & Christoph Adami[1,2,3,4]

Most mutations are deleterious and cause a reduction in population fitness known as the mutational load. In small populations, weakened selection against slightly-deleterious mutations results in an additional fitness reduction. Many studies have established that populations can evolve a reduced mutational load by evolving mutational robustness, but it is uncertain whether small populations can evolve a reduced susceptibility to drift-related fitness declines. Here, using mathematical modeling and digital experimental evolution, we show that small populations do evolve a reduced vulnerability to drift, or 'drift robustness'. We find that, compared to genotypes from large populations, genotypes from small populations have a decreased likelihood of small-effect deleterious mutations, thus causing small-population genotypes to be drift-robust. We further show that drift robustness is not adaptive, but instead arises because small populations can only maintain fitness on drift-robust fitness peaks. These results have implications for genome evolution in organisms with small effective population sizes.

[1] Department of Microbiology & Molecular Genetics, Michigan State University, East Lansing, MI 48824, USA. [2] BEACON Center for the Study of Evolution in Action, Michigan State University, East Lansing, MI 48824, USA. [3] Program in Ecology, Evolutionary Biology, and Behavior, Michigan State University, East Lansing, MI 48824, USA. [4] Department of Physics and Astronomy, Michigan State University, East Lansing, MI 48824, USA. Correspondence and requests for materials should be addressed to C.A. (email: adami@msu.edu)

One consequence of the power of adaptation is that the majority of mutations reduce their bearer's fitness[1]. The recurring nature of these deleterious mutations results in an equilibrium reduction of population fitness at mutation-selection balance. At the population level, this reduction in fitness is known as the genetic or mutational load[2–5]. As selection generally acts to increase a population's mean fitness, one avenue for selection to increase mean fitness is to reduce the mutational load by altering mutation-selection balance and increasing mutational robustness[6, 7]. The evolution of mutational robustness has been demonstrated using theoretical modeling[8–11], digital experimental evolution[12–14], and microbial experimental evolution[15–17].

Recurring deleterious mutations are not the only strain on fitness. In small populations, genetic drift leads to the fixation of slightly-deleterious mutations that bring about a reduction in fitness[18, 19]. Over time, genetic drift can lead to continual fitness declines and ultimately population extinction[20, 21]. In asexual populations, this phenomenon of fitness decline is known as Muller's ratchet[22] and is thought to play a role in the evolution of mitochondria[23], bacterial endosymbionts[24], the Y chromosome[25], and other obligate asexual lineages. Muller's ratchet may explain why there are few long-lived obligate asexual species and may provide a selection pressure for the evolution of sexual recombination[26]. However, it was recently proposed that small populations do not continuously decline in fitness, but only do so until they reach drift-selection balance when the fixation of beneficial mutations counteracts the fixation of slightly-deleterious mutations[18, 19, 27, 28]. Furthermore, Muller's ratchet may be limited in strength if small populations can alter drift-selection balance and evolve drift robustness. However, it is unknown if populations can evolve drift robustness, or what genetic and evolutionary mechanisms could cause drift robustness.

Here, we propose a hypothesis concerning the evolution of drift robustness in small populations. Consider evolution on a single-peak fitness landscape (Fig. 1a). In a large population (defined here such that its effective population size is larger than the inverse of every selection coefficient in the landscape), natural selection will ultimately lead to the fixation of all beneficial mutations. In a small population, while selection may also lead to the fixation of these beneficial mutations, weakened purifying selection inherent to small populations will result in the subsequent *loss* of these beneficial mutations. Thus, while a large population can maintain itself at the top of the fitness peak, a

small population is unable to maintain fitness due to an increased rate of fixation of slightly-deleterious mutations. Therefore, this small population will not occupy the top of the fitness peak, but some lower area where the fixation of slightly-beneficial mutations and the fixation of slightly-deleterious mutations balance out[28].

Now, consider a fitness landscape with two fitness peaks, with one peak slightly higher than the other peak (Fig. 1b). We will denote the higher peak as the "drift-fragile" fitness peak. A population evolves towards this peak by fixing a sequence of small-effect beneficial mutations. As a consequence, the genotype at the top of the peak will have many small-effect deleterious mutations in its mutational neighborhood. We will denote the lower peak as the "drift-robust" fitness peak. A population evolves to this peak by fixing a sequence of large-effect beneficial mutations and the genotype at the top of the peak will have many large-effect deleterious mutations in its mutational neighborhood. The question is: how will small and large populations evolve on this fitness landscape?

According to our hypothesis, large populations will evolve towards the drift-fragile fitness peak and small populations will evolve towards the drift-robust fitness peak. This hypothesis is similar to the idea of the "Survival of the Flattest" effect, where mutationally-robust genotypes will out-compete fitter, but more mutationally-fragile genotypes at high mutation rates[12]. However, we stress that the evolutionary mechanism behind this trend is not the out-competition of drift-fragile genotypes by drift-robust genotypes in small populations. Instead, we propose that small populations evolve to drift-robust fitness peaks because these populations can only maintain themselves on drift-robust areas of the fitness landscape. If a small population would evolve towards a drift-fragile part of the fitness landscape, it would subsequently fix deleterious mutations and decrease in fitness until it evolved back to a drift-robust area. Large populations can easily maintain fitness in drift-fragile areas, and thus we expect them to evolve to the higher fitness peak.

Here, we demonstrate that small populations should evolve drift robustness in accordance with our hypothesis. We first confirm the logic behind this hypothesis with a two-peak fitness landscape mathematical model and show that drift robustness will evolve in small, but not large, populations in a fitness landscape with a drift-fragile fitness peak and a lower-fitness drift-robust fitness peak. Then, we use digital experimental evolution with the Avida system[29] to test this hypothesis in a complex

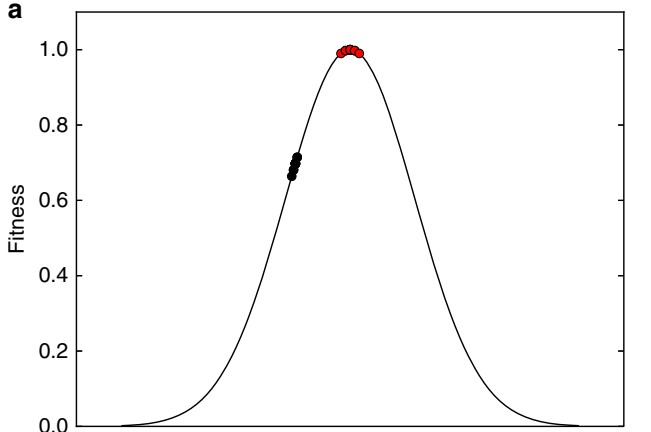
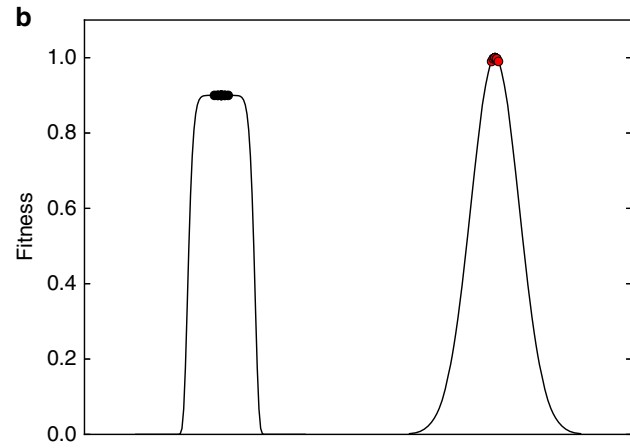

**Fig. 1** Conceptual diagram of drift robustness. **a** A single-peak fitness landscape. In this landscape, the large population (*red circles*) can climb to the top of the fitness peak, while the small population (*black circles*) can only maintain fitness on an intermediate part of the peak. **b** A multi-peak fitness landscape. The large population evolves to the same fitness peak as in **a**. The small population evolves to the steeper, drift-robust peak. While this peak is still lower than the drift-fragile peak, the small population attains greater fitness than it would have on the drift-fragile peak in the single-peak fitness landscape

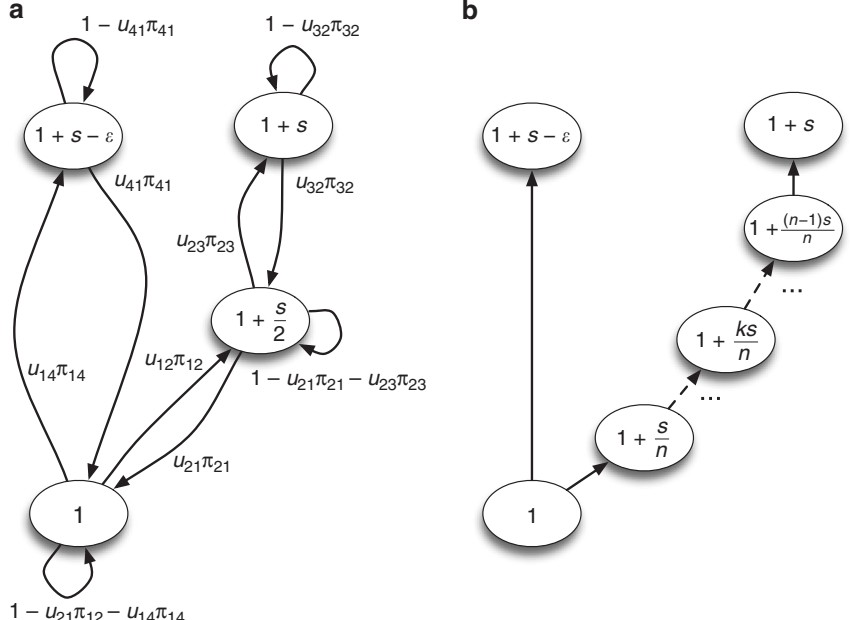

**Fig. 2** The fitness landscapes for the Markov model to test for the evolution of drift robustness. Each circle represents one genotype and is labeled with its fitness. Each arrow represents the transition between one genotype to another (including the identical genotype) and is labeled with the transition probability. **a** The fitness landscape for the minimal model. $s$ represents the selection coefficient of the drift-fragile peak and $\epsilon$ represents the small fitness difference between the drift-fragile peak and the drift-robust peak. $u_{ij}$ and $\pi_{ij}$ represent the mutation rate between genotypes and probability of fixation from one genotype to another, respectively. **b** The fitness landscape for the extended model. Variables as in **a**. Transition probabilities omitted for clarity

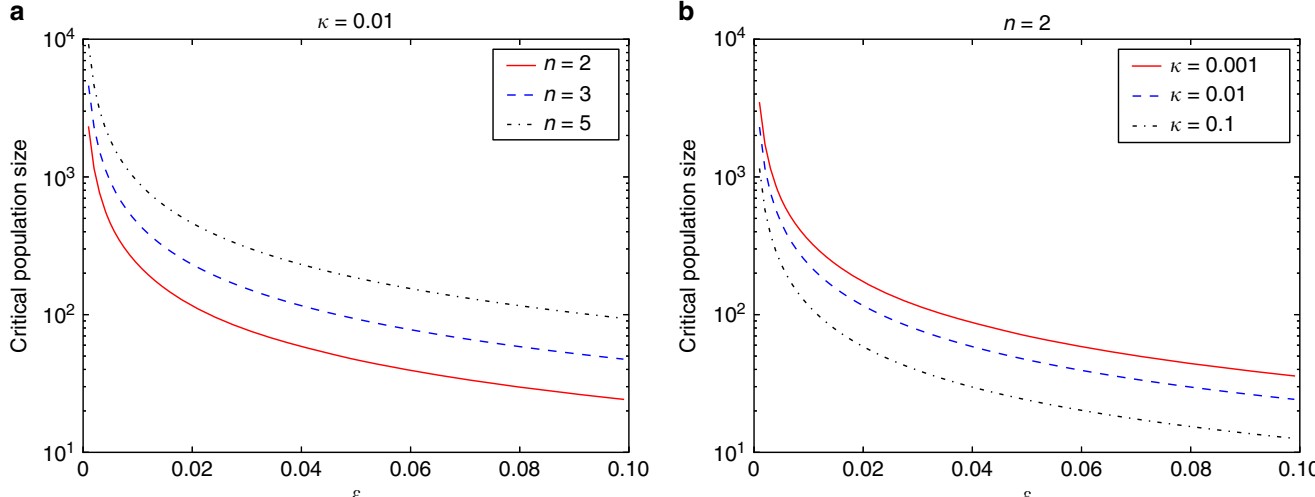

**Fig. 3** Critical population size for shift between robust and fragile peaks. **a** Results for various $n$ values with $\kappa = 0.01$. **b** Results for various $\kappa$ values with $n = 2$

fitness landscape. We find that small populations of digital organisms evolve towards fitness peaks with a low likelihood of slightly-deleterious mutations, while large populations evolve towards fitness peaks with a high likelihood of slightly-deleterious mutations. We end by discussing the implications of these results for organisms exposed to strong genetic drift, including bacterial endosymbionts and RNA viruses.

## Results

**A mathematical model of drift robustness.** To test our drift robustness hypothesis, we designed a minimal mathematical model in order to test the conditions under which drift robustness will evolve in small populations, while drift fragility will evolve in large populations (see Methods). We designed a fitness landscape

with a wild-type genotype with fitness $w_1 = 1$ and two fitness peaks with $w_3 = 1 + s$ and $w_4 = 1 + s - \epsilon$, respectively (Fig. 2a), where $s$ is a fitness advantage (in percent), and $\epsilon$ quantifies how much lower the fitness of the peak $w_4$ is compared to $w_3$. The lower of these peaks is the drift-robust fitness peak, as it can be reached from the wild-type genotype by fixing a strongly-beneficial mutation of size $s - \epsilon$. As a consequence, this peak's mutational neighborhood only consists of strongly-deleterious mutations. The drift-fragile peak is, in contrast, reached by first fixing an intermediate genotype with fitness $w_2 = 1 + \frac{s}{2}$ and then fixing another mutation with the same fitness effect. Both these mutants are slightly-beneficial and thus the drift-fragile peak will have a mutational neighborhood of slightly-deleterious mutations. In an extended model (Fig. 2b), the drift-fragile peak has $n - 1$ intermediate steps that are reached with mutations of step size

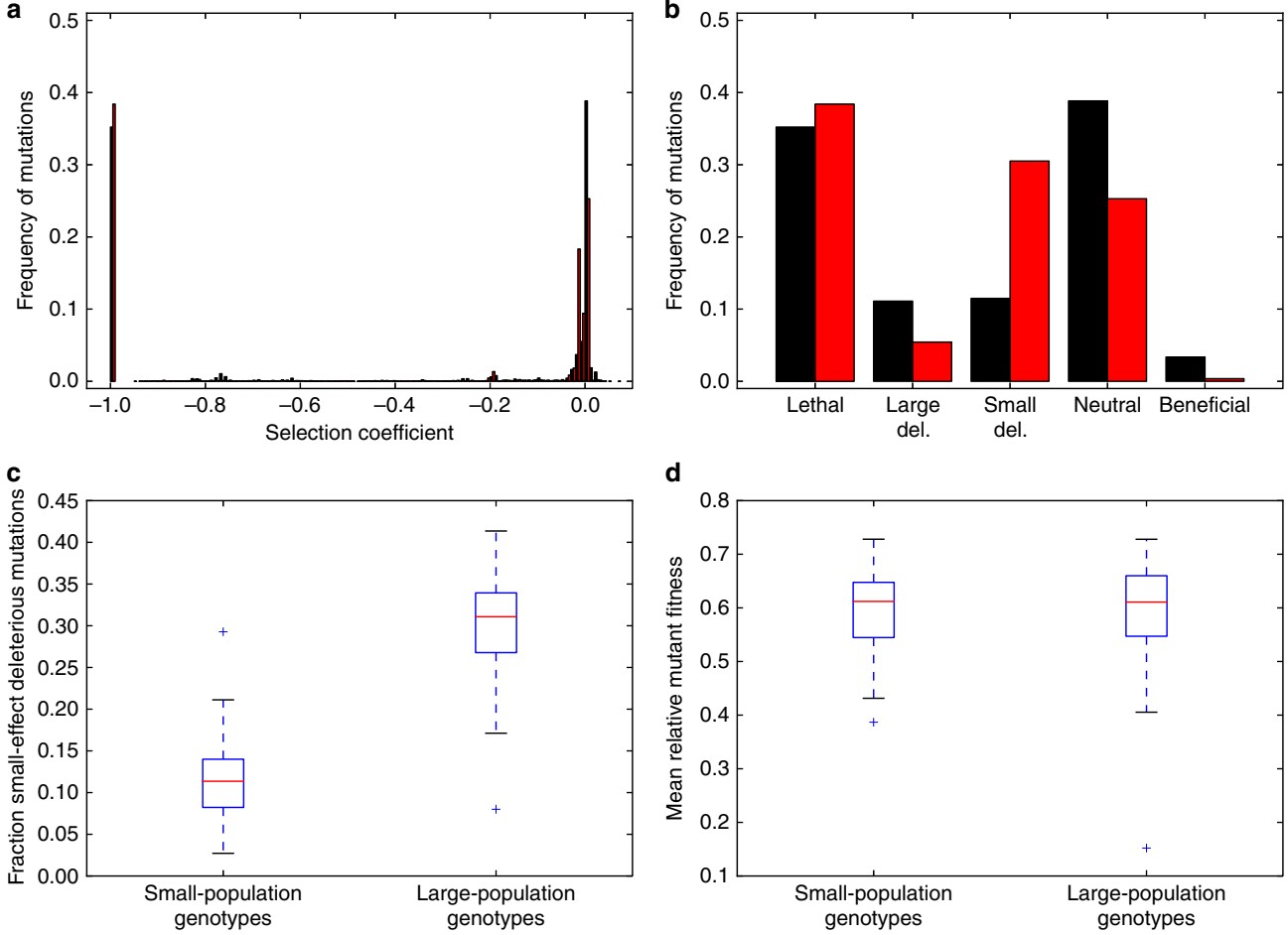

**Fig. 4** Differences in mutational effects between small-population genotypes and large-population genotypes. Black and red represent small-population and large-population genotypes, respectively. **a** The combined distribution of fitness effects (DFE) across all 100 small-population genotypes and 100 large-population genotypes. **b** Same data as in panel a, but grouped into different classes of mutations, where a small-effect deleterious mutation is defined as having an effect less than 5%. Here, and throughout, deleterious mutation refers to viable deleterious mutations, while lethal mutation refers to non-viable deleterious mutations. **c** The likelihood of a small-effect deleterious mutation for small-population and large-population genotypes. Red lines are medians, edges of the box are first and third quartile, whiskers are at most 1.5 times the interquartile range, and the plus signs are outliers. **d** The mean relative fitness of every possible point mutation (1250 mutations) for each genotype

$s/n$, so that choosing $n$ allows us to vary the steepness of the slope of the drift-fragile peak.

When disregarding landscape structure, population genetics arguments imply that large populations will fix on the higher of the two peaks, while if the population size is small, the difference in fitness between peaks is irrelevant and a population should fix on either peak with approximately equal probability. Instead, our model predicts that when deleterious mutations are more abundant than beneficial mutations, there is a broad range of parameter values where the small populations evolve to predominantly fix at the drift-robust fitness peak (even though it is of lower fitness) and the large populations evolve to the drift-fragile fitness peak (Fig. 3). For the minimal model, we derive a critical population size at which the small population shifts from fixing at the higher peak to the lower one instead, with the assumption that beneficial mutations are exponentially-distributed, (see Methods)

$$N_{crit} = 1 + \frac{\log \kappa^{-1}}{2\epsilon} \quad (1)$$

where $\kappa = \frac{u_b}{\bar{s}} < 1$ is the ratio between the beneficial mutation rate $u_b$ and the mean beneficial fitness effect $\bar{s}$.

In the extended model, where $n$ (and thus the slope of the drift-fragile peak) can vary, we find the critical population size to be:

$$N_{crit} = 1 + (n-1) \frac{\log \kappa^{-1}}{2\epsilon} \quad (2)$$

where $n$ is the number of mutations required to reach the top of the drift fragile peak. This general equation makes the following predictions concerning how populations should shift from drift-fragile peaks to drift-robust peaks. As the fitness deficit of the drift-robust fitness peak increases ($\epsilon$), the critical population size, and thus range of population sizes that lead to the evolution of drift robustness, also decrease (Fig. 3). In other words, small populations will only preferentially evolve towards the drift-robust fitness peak if the trade-off between drift robustness and fitness is not too severe. If the drift-robust peak results in extremely low fitness, the small population will evolve as far up the drift-fragile peak as it can while maintaining fitness. As the shallowness of the slope of the drift-fragile peak increases (i.e., $n$, or the number of mutations to reach the drift-fragile peak, increases), the critical population size also increases. This result argues that the range of population sizes leading to the evolution of drift robustness is greater as the mutations that lead to the drift-fragile peak are more frequent, with a decreased beneficial

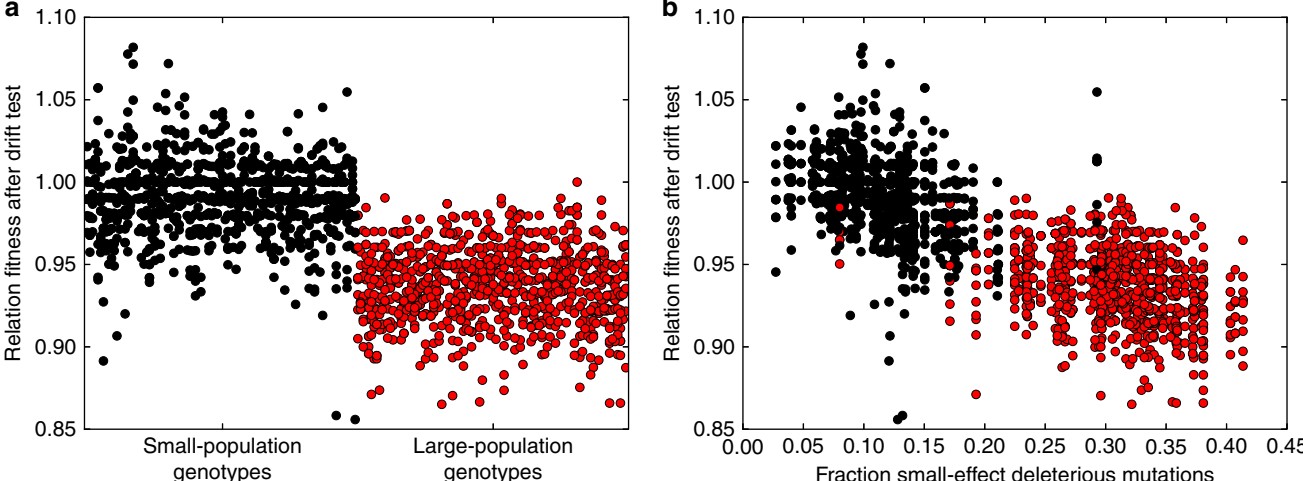

**Fig. 5** Small-population genotypes are drift-robust due to a decreased likelihood of small-effect deleterious mutations. Black and red data points represent small-population genotypes and large-population genotypes, respectively. **a** Relative fitness of the most-abundant genotype from every population during the drift robustness test. Each circle represents the relative fitness of one genotype from one replicate. **b** Relationship between relative fitness in the drift robustness test (**a**) and the likelihood of small-effect deleterious mutations (Fig. 4c)

effect. Finally, as $\kappa$ decreases [either by a decrease in the beneficial mutation rate ($u_b$) or an increase in the height of the fitness peaks ($s$)] the critical population size increases, demonstrating that the larger the differential between the flux of beneficial mutations towards the peaks, the larger the critical population size. We should note here that $\kappa$ has a weaker influence on $N_{\text{crit}}$ than $\epsilon$ or $n$. This lesser influence, due to the equation containing $\log k^{-1}$, exists because there is only a slight difference in the fixation probability of beneficial mutations between small and large populations. The relevant difference comes down to the lack of maintainability of these beneficial mutations in small populations, an effect captured by $n$, the number of beneficial mutations required to reach the drift-fragile peak.

**Drift robustness in digital organisms**. The mathematical model supports our hypothesis for the evolution of drift robustness in small populations, but it rests on a number of assumptions that may alter the evolution of drift robustness in complex fitness landscapes. For instance, we assumed that populations can be viewed as monomorphic and evolution proceeds as transitions from one genotype to another (these models are broadly known as origin-fixation models[30]). We also used a fitness landscape with only two fitness peaks, while biological fitness landscapes certainly contain many fitness peaks.

To test if small populations evolve drift robustness in a complex fitness landscape, we used the digital evolution system Avida[29]. In Avida, a population of self-replicating computer programs ("avidians") compete for the memory space and CPU time necessary for reproduction. During self-replication, random mutations occur, potentially altering the new avidian's reproduction speed. When an avidian successfully reproduces, its offspring replaces a random individual in the population, resulting in genetic drift. As avidians that replicate faster will produce more offspring per unit time than avidians with slower replication speeds, faster replicators are selected for and spread mutations that enable faster replication. Because Avida populations undergo selection, mutation, and drift, they represent a digital model system to study fundamental questions concerning evolutionary dynamics.

To test for the evolution of drift robustness in small Avidian populations, we evolved 100 replicate populations at small ($10^2$ individuals) and 100 populations at large ($10^4$ individuals)

population sizes for $10^5$ generations. From each population, we isolated the most abundant genotype at the end of the experiment; we will refer to these genotypes as the 100 small-population genotypes and the 100 large-population genotypes. Small populations evolved a lower relative fitness than large populations (median = 1.85 vs. median = 2.05, Mann Whitney $U$ = 2237.0, $n$ = 100, $p$ = $7.31 \times 10^{-12}$ one-tailed), as expected for populations that experience a smaller beneficial mutation supply over the course of the experiment.

**Small populations evolve an altered DFE**. To look for signs of drift robustness, we studied differences in the Distribution of Fitness Effects (DFE) of de-novo mutations for small-population genotypes and large-population genotypes. First, we generated every possible point mutation for all genotypes and combined these data into one DFE (Fig. 4a). Both show the typical properties of DFE's found in biological organisms: most mutations are either lethal or have little effect[1]. However, there are some differences. Small-population genotypes have an excess of neutral, beneficial, and strongly deleterious mutations (defined as viable deleterious mutations with a fitness effect greater than or equal to 5%; Fig. 4b), while large-population genotypes have an excess of small-effect deleterious mutations (defined as viable deleterious mutations with a fitness effect less than 5%; Fig. 4b). We confirmed that these trends hold when we calculated a DFE for each genotype (rather than one DFE for all genotypes from a given population size) as follows. Small population genotypes had a greater likelihood of beneficial mutations (median = 0.0256 vs. median = 0.0008, Mann Whitney $U$ = 413.5, $n$ = 100, $p$ = $6.26 \times 10^{-30}$ one-tailed), neutral mutations (median = 0.40 vs. median = 0.26, Mann Whitney $U$ = 321.0, $n$ = 100, $p$ = $1.45 \times 10^{-30}$ one-tailed), large-effect deleterious mutations (median = 0.084 vs. median = 0.054, Mann Whitney $U$ = 2854.0, $n$ = 100, $p$ = $7.90 \times 10^{-8}$ one-tailed), and a lesser likelihood of lethal mutations (median = 0.33 vs. median = 0.37, Mann Whitney $U$ = 4031.5, $n$ = 100, $p$ = $9.00 \times 10^{-3}$ one-tailed) and small-effect deleterious mutations (median = 0.11 vs. median = 0.31, Mann Whitney $U$ = 124.5, $n$ = 100, $p$ = $5.13 \times 10^{-33}$ one-tailed; Fig. 4c). Additionally, there was no difference in the average single-mutant relative fitness for small-population genotypes and large-population genotypes, even though there were fitness differences between the population-size treatments (median

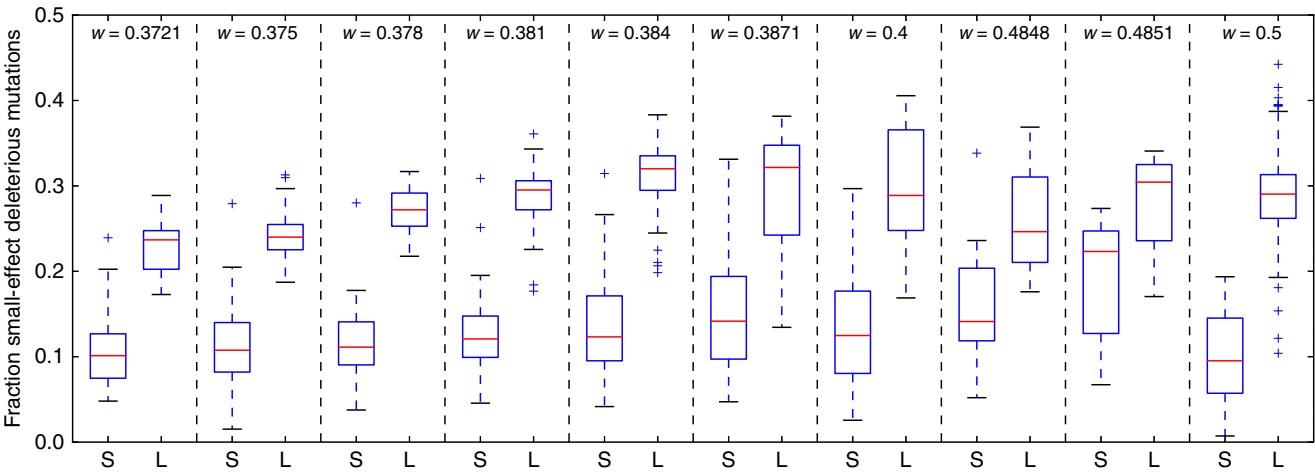

**Fig. 6** Fraction of small-effect deleterious mutations for genotypes from small populations and large populations with equal fitness. Each area separated by a dashed line shows genotypes with equal fitness ($w$). S and L represent small-population and large-population genotypes, respectively. Differences for each fitness value are significant. (Mann-Whitney U-test; Bonferroni-corrected $p < 3 \times 10^{-3}$). Box plots as described for Fig. 4c

$w = 0.612$ vs. median $w = 0.611$, Mann Whitney $U = 4890.0$, $n = 100$, $p = 0.39$ one-tailed; Fig. 4d).

**Small population genotypes are drift-robust**. The lack of small-effect deleterious mutations in small populations suggests that these populations adapted to drift-robust fitness peaks and that the large populations adapted to drift-fragile peaks. To test if these small-population genotypes are drift-robust, we took the 100 small-population genotypes and 100 large-population genotypes and measured these genotypes' change in fitness when placed in an environment with strong genetic drift (i.e., low population size). We evolved 10 populations of 50 individuals for each genotype for $10^3$ generations and measured their change in fitness. Small-population genotypes clearly declined less in fitness than large-population genotypes (median decline = 1% vs. median decline = 6%; Mann Whitney $U = 43959.5$, $n = 1000$, $p = 1.44 \times 10^{-273}$ one-tailed; Fig. 5a). Furthermore, a genotype's decline in fitness is correlated with its likelihood of a small-effect deleterious mutation, supporting the idea that small populations have evolved to fitness peaks with a low likelihood of small-effect deleterious mutations due to the peak's drift robustness (Spearman's $\rho = 0.80$, $p \approx 0$; Fig. 5b).

**Drift robustness is not due to fitness differences**. The above results are consistent with the hypothesis that small populations evolve to drift-robust fitness peaks and large populations evolve to drift-fragile fitness peaks. However, one could argue the results are also consistent with evolution on a single-peaked fitness landscape (Fig. 1a). The small populations we examined have lower fitness than the large populations and thus could have a decreased likelihood of small-effect deleterious mutations and more robustness to drift because they are further down on the fitness peak and cannot climb the rest of the peak. To test if our results were due to the lower fitness of the small-population genotypes, we isolated genotypes of the same fitness from the evolutionary lineages of the small and large populations (see Methods for details). We then compared these genotypes' likelihood of small-effect deleterious mutations. Genotypes from the small populations had a decreased likelihood of small-effect deleterious mutations compared to genotypes from large populations for every examined fitness value (Fig. 6). These results support the hypothesis that small populations have evolved to different fitness peaks than large populations and are not merely occupying a lower region of the same fitness peak.

**Epistatic mutations lead to drift robustness**. Next, we examined the mutations that enabled small populations to evolve towards drift-robust peaks. Our mathematical model has a drift-robust peak accessible by strongly-beneficial mutations and a drift-fragile peak accessible by slightly-beneficial mutations. Therefore, we first examined the distribution of fitness effects for maintained beneficial mutations (beneficial mutations whose fitness gain was at-least partially maintained during subsequent evolution) to see if small populations fixed more strongly-beneficial mutations (Fig. 7a). While small populations did fix a significantly large proportion of maintained strongly-beneficial ($s > 0.05$) mutations (median = 0.06 vs. median = 0.05, Mann Whitney $U = 4067.5$, $n = 100$, $p = 0.01$ one-tailed), the difference was slight.

The fixation of strongly-beneficial mutations is not the only way small populations could climb drift-robust peaks. Small populations could also climb drift-robust fitness peaks through epistatic beneficial mutations that decreased the likelihood of small-effect deleterious mutations. By decreasing the likelihood of small-effect deleterious mutations, the likelihood that a future small-effect deleterious mutation will arise and fix is also decreased. Thus, these epistatic beneficial mutations can be maintained by small populations.

To see if these types of mutations were fixed in the small populations, we first looked at the correlation between the fitness of maintained beneficial mutations and their genotypes' likelihood of deleterious mutations for each population. In a non-epistatic fitness landscape, we would expect the likelihood of deleterious mutations to increase as fitness increases due to the fixation of, and the subsequent decrease in the likelihood of, beneficial mutations. However, in some epistatic fitness landscapes, this correlation is not guaranteed to exist, as the fixation of beneficial mutations may alter the fitness effects of mutations at other loci. In fact, small populations showed a significant decrease in the correlation between fitness and deleterious mutational likelihood when compared to large populations (median Spearman's $\rho = 0.24$ vs. median Spearman's $\rho = 0.73$, Mann Whitney $U = 2082.0$, $n = 100$, $p = 5.07 \times 10^{-13}$ one-tailed; Fig. 7b). This result is consistent with small populations evolving towards fitness peaks with a decreased likelihood of deleterious mutations through the fixation of epistatic mutations.

Next, we looked for specific mutational signatures of the fixation of epistatic beneficial mutations in small populations. We found that 22 small populations had fixed beneficial mutations

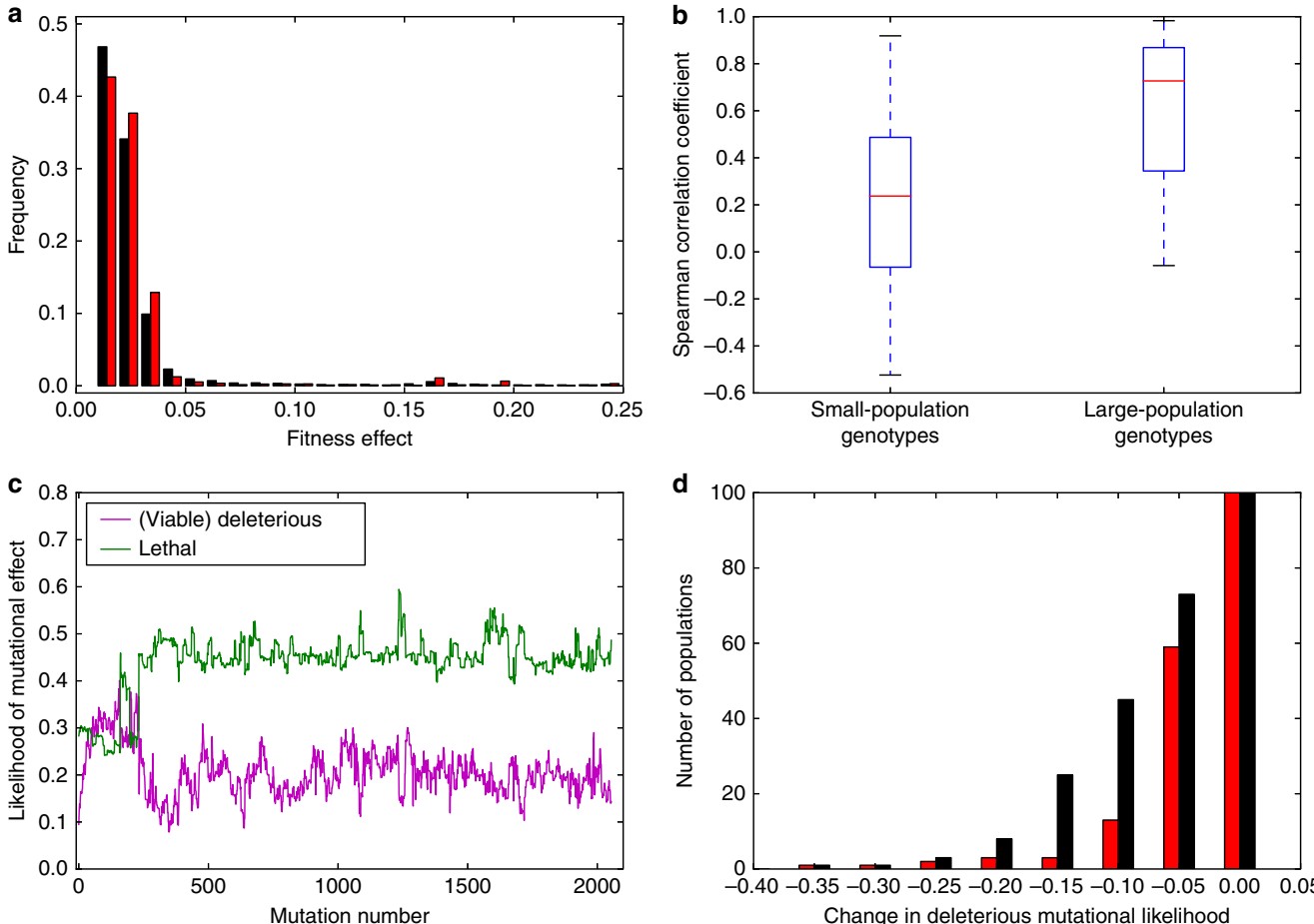

**Fig. 7** Evidence of small-population adaptation to drift-robust fitness peaks. **a** Distribution of maintained beneficial mutational effects for (effects for mutations whose fitness gain was at-least partially maintained during subsequent evolution) small-population genotypes (*black*) and large-population genotypes (*red*). **b** Spearman correlation coefficients between fitness and the likelihood of a deleterious mutation for each maintained beneficial mutation from each population **c** The likelihood of (viable) deleterious and lethal mutations, shown in magenta and green, respectively, in a representative small population's lineage. The strong decrease in the likelihood of a (viable) deleterious mutation early in the population's history is evidence of epistatic mutations resulting in drift robustness. **d** Number of populations that fixed a maintained beneficial mutation that decreased the likelihood of a (viable) deleterious mutation by at least a specified amount. Colors as in **a**

that reduced the likelihood of a deleterious mutation by 50%, while only 4 large populations did so. All of these mutations increased the likelihood of lethal mutations (mean increase = 72%, 2 × S.E. = 11%; Fig. 7c). We then studied the magnitude of the decrease in the likelihood of deleterious mutations. Forty-five small populations fixed mutations that decreased this deleterious likelihood by at least 0.1, while only 13 large populations did so (Fig. 7d). All of these mutations also increased the likelihood of lethal mutations. Finally, we confirmed that these epistatic mutations specifically decreased the likelihood of small-effect deleterious mutations. Most of the decrease in the likelihood of deleterious mutations consisted of a decrease in small-effect deleterious mutations (median percentage of decrease = 83.9%, interquartile range = 20.6–97.1%), further suggesting that small populations evolve drift robustness by fixing beneficial mutations that decrease the likelihood of small-effect deleterious mutations and increase the likelihood of lethal mutations.

**Deleterious mutations drive the evolution of drift robustness.** Finally, to test whether drift robustness evolves in small populations because these populations can only maintain fitness in drift-robust areas of the fitness landscape, we performed further

evolution experiments where deleterious mutations were prevented from occurring (see Methods for further details). In this environment, populations cannot decline in fitness, so small populations do not maintain fitness differently on drift-robust and drift-fragile fitness peaks. We evolved 100 small populations without deleterious mutations under the main experimental conditions. Small population genotypes evolved greater relative fitness without deleterious mutations than in the treatment with deleterious mutations (median = 2.05 vs. median = 1.85, Mann Whitney $U = 2464.5$, $n = 100$, $p = 2.92 \times 10^{-10}$ one-tailed; Fig. 8a). As expected in an environment where fitness maintenance was not a factor, small-population genotypes had a greater likelihood of small-effect deleterious mutations (median = 0.19 vs. median = 0.11, Mann Whitney $U = 1769.0$, $n = 100$, $p = 1.47 \times 10^{-15}$ one-tailed; Fig. 8b). These small-population genotypes were less robust to genetic drift (median fitness decline of 5% vs. median fitness decline of 1%, Mann Whitney $U = 118333.0$, $n = 100$, $p = 2.25 \times 10^{-192}$ one-tailed; Fig. 8c) and this decreased robustness correlates with their increased frequency of small-effect deleterious mutations (Spearman's $\rho = -0.43$, $p = 2.07 \times 10^{-45}$; Fig. 8d). These results suggest that small populations evolve to alternative areas of the fitness landscape if they can maintain small-effect beneficial mutations.

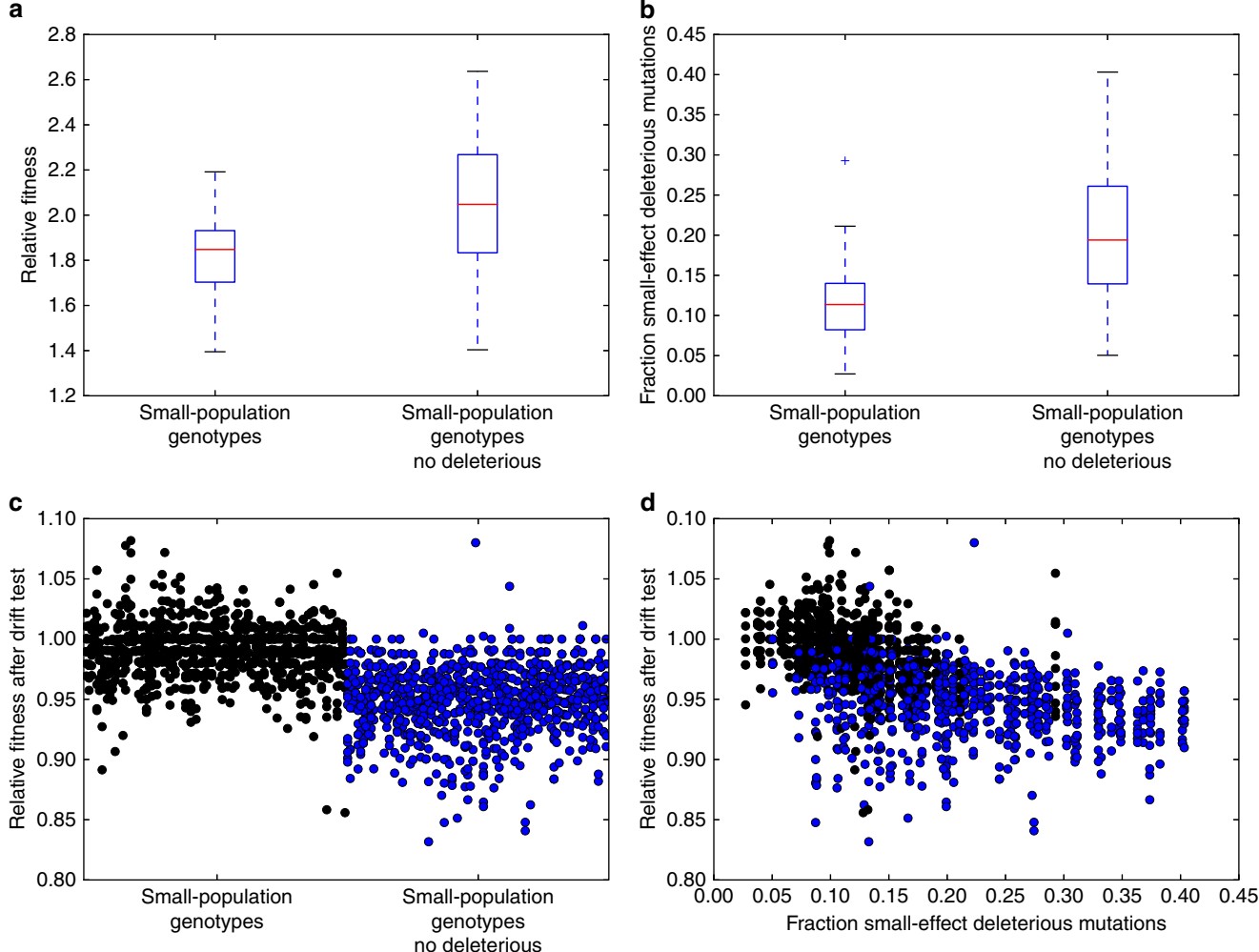

**Fig. 8** The evolution of drift robustness in small populations with or without deleterious mutations in the initial adaptation experiments. Black and blue data points represent small-population genotypes adapted with deleterious mutations and small-population genotypes adapted without deleterious mutations, respectively. **a** Relative fitness to the ancestral genotype after $10^5$ generations of adaptation. Box plots as described for Fig. 4c. **b** Likelihood of small-effect deleterious mutations. **c** Relative fitness of the most-abundant genotype from every population during the drift robustness test. Each circle represents the relative fitness of one genotype from one replicate. **d** Relationship between relative fitness in the drift robustness test (**b**) and the likelihood of small-effect deleterious mutations (**c**)

## Discussion

Our results suggest the following explanation for the evolution of drift robustness in small populations. Small populations cannot adapt to fitness peaks with a high likelihood of small-effect deleterious mutations. If small populations climb these peaks, genetic drift will cause them to lose previously-fixed beneficial mutations, leading to a decrease in fitness. In other words, small populations cannot maintain themselves on drift-fragile fitness peaks. Thus, small populations, if they do adapt, must adapt to drift-robust fitness peaks. Due to the relative increased strength of selection, large populations do not face this constraint and adapt to drift-fragile peaks. Therefore, our results argue that small populations and large populations should evolve to different areas of the fitness landscape and evolve qualitatively-different genetic architecture.

We should emphasize here that there are certain requirements for the evolution of drift robustness in small populations. First, the fitness landscape must contain multiple peaks; some peaks must be drift-robust with few small-effect deleterious mutations and some must be drift-fragile with many small-effect deleterious mutations. If there is only one fitness peak, small populations would still likely have a decreased likelihood of small-effect

deleterious mutations. However, this would occur because these populations have failed to maintain small-effect beneficial mutations, not because they have evolved to drift-robust peaks. Second, the requirement of multiple fitness peaks further implies that this effect will only be seen in fitness landscapes with strong epistasis in parts of the landscape, as (sign) epistasis leads to multiple fitness peaks[31]. Third, there must be evolutionary trajectories between drift-robust fitness peaks and drift-fragile fitness peaks. Otherwise, small populations would only evolve downwards on a drift-fragile fitness peak. Finally, there must be more trajectories to drift-fragile fitness peaks than drift-robust fitness peaks.

We are not the first to propose that small populations will evolve robustness mechanisms in response to their deleterious mutational burden. However, these mechanisms are usually discussed in terms of mutational robustness, not robustness to drift. Previous studies provided two characteristics of the evolution of mutational robustness in small populations. First, small populations should preferentially evolve to lower fitness peaks with more "redundancy," defined as a decreased average deleterious mutational effect and large populations should evolve to fitness peaks with a high average deleterious mutational effect[9, 14]. Our results

suggest opposite evolutionary trajectories for small and large populations. While our results concerning the evolution of drift robustness do suggest that small populations evolve to lower fitness peaks, and small populations do evolve more redundancy in terms of exactly-neutral mutations, these small populations do not evolve towards fitness peaks with a decreased deleterious mutational effect (Fig. 4d). In fact, they evolve towards fitness peaks with a minimal likelihood of small-effect deleterious mutations. This discrepancy likely exists due to the fitness landscape used to study the evolution of redundancy in small populations: the mutations in that fitness landscape were all small-effect deleterious mutations[9]. Thus, small populations could not maintain fitness except on the flattest of fitness landscapes[9]. In a version of this model with multiple fitness peaks (e.g., one with small-effect deleterious mutations and one with large-effect deleterious mutations), we expect that small populations would evolve to the peak with large-effect deleterious mutations. Such an outcome was recently predicted for populations evolving at very high mutation rates[32], although a different model predicts that small populations should evolve to areas that minimize the deleterious effect of mutations[33], in accordance with Krakauer and Plotkin's model[9].

The second characteristic of mutational robustness in small populations is that these populations should evolve "global" robustness mechanisms, such as error-correction mechanisms, that affect many loci[10, 11, 34]. There are no global error-correction mechanisms available to the avidian genomes here (although one could allow the evolution of mutation rates, e.g.,[35]). However, we did find that small populations preferentially fixed epistatic mutations that strongly altered the likelihood of deleterious mutations (Fig. 6c, d). These mutations are global in the sense that they alter the fitness effects of mutations at multiple loci. However, unlike previous work that suggested small populations should fix global solutions that reduce the effect of deleterious mutations[10, 11, 34], we found that these mutations increased the likelihood of lethal mutations. We do not have strong evidence that this increased lethality is essential and expect that small populations could also fix mutations that increased the likelihood of neutral mutations while reducing the likelihood of deleterious mutations if they exist in the Avida fitness landscape. Generally, our results emphasize that the evolutionary process behind drift robustness is the trend to reduce the likelihood of small-effect deleterious mutations, which can be achieved in multiple ways.

As the evolution of drift robustness relies on a number of conditions, we may ask which empirical fitness landscapes, or which organisms, meet these criteria? Candidates for organisms with drift-robust genomes include those that undergo severe bottlenecks during their lifecycle, including bacterial endosymbionts[24] and RNA viruses[36]. There is evidence that both bacterial endosymbionts[17, 37–39] and RNA viruses[40, 41] have evolved alternate genome architectures in response to their population-genetic environment. In endosymbionts, drift robustness could be achieved by choosing rare codons in such a way that substitutions are highly deleterious, and indeed proteins in *Buchnera* have been found to be exceptionally resistant to drift[42]. However, there has been to date no systematic study of how different organisms respond to strong genetic drift. Future work with biological organisms should establish the circumstances that cause organisms to vary in their robustness to genetic drift. Furthermore, experimental evolution may be able to produce organisms with drift-robust genomes whose architecture can be studied directly.

## Methods
**Mathematical model of drift robustness**. We describe a model to study the minimal conditions required for the evolution of drift robustness. Based on our

hypothesis, we need to study evolution on a fitness landscape with at least two fitness peaks: one drift-robust peak with few small-effect deleterious mutations and one drift-fragile peak with many small-effect deleterious mutations. We assume that deleterious mutations are more frequent than beneficial mutations and that beneficial mutations of large-effect are less frequent than beneficial mutations of small-effect. Finally, there must also be a mutational path between the drift-fragile peak and the drift-robust peak. Drift robustness on such a landscape would manifest itself when a population that predominantly occupies a high (drift-fragile) fitness peak when under selection at large population sizes, switches instead to the lower (drift-robust) fitness peak when the population is small. Below we will calculate the critical population size at which this switch occurs.

We design a fitness landscape with four genotypes, represented by four nodes (Fig. 2a). Genotype 1 (the wild-type) has fitness $w_1 = 1$ and is the ancestral genotype for our populations. Genotypes 2 and 3, with fitness $w_2 = 1 + \frac{s}{2}$ and $w_3 = 1 + s$, respectively represent the genotypes on the drift-fragile fitness peak ($s$ is the size of the fitness benefit). Genotype 4, with fitness $w_4 = 1 + s - \epsilon$, illustrates the drift-robust fitness peak at lower fitness (lower by $\epsilon > 0$). In the extended version of this model that we present later, we discuss the case where an arbitrary number of mutations lie "on the path" towards the drift-fragile peak (thus increasing the peak's fragility).

The likelihood that a mutation on the genetic background of genotype $i$ leads to genotype $j$ is denoted by $u_{ij}$ and the probability of fixation of that mutation is denoted by $\pi_{ij}$, with $0 < u_{ij}, \pi_{ij} \leq 1$. Therefore, the probability the population will evolve from genotype $i$ to genotype $j$ is $u_{ij}\pi_{ij}$ and the probability the population will not change is $1 - \sum_j u_{ij}\pi_{ij}$. Mutations cannot occur from the drift-fragile peak to the drift-robust peak and vice-versa, but an indirect path between them exists. To allow for this dynamic, back-mutations can occur.

We assume that evolution occurs in a mutation-limited environment (weak mutation, strong selection limit), where the population is almost always monoclonal. When a mutation arises, it will either go extinct or takeover the population. This assumption allows us to treat evolution as a Markov chain[43]. We then solve for the stationary distribution of mutants in the population, to calculate the likelihood a population with defined characteristics will evolve to either one fitness peak or the other.

To solve the Markov chain, we first write down the transition matrix $T$ as:

$$T = \begin{bmatrix} 1 - u_{12}\pi_{12} - u_{14}\pi_{14} & u_{12}\pi_{12} & 0 & u_{14}\pi_{14} \\ u_{21}\pi_{21} & 1 - u_{21}\pi_{21} - u_{23}\pi_{23} & u_{23}\pi_{23} & 0 \\ 0 & u_{32}\pi_{32} & 1 - u_{23}\pi_{23} & 0 \\ u_{41}\pi_{41} & 0 & 0 & 1 - u_{41}\pi_{41} \end{bmatrix}.$$

$$(3)$$

The stationary distribution $\vec{x}^* = \left(x_1^*, x_2^*, x_3^*, x_4^*\right)$ is the left eigenvector of the transition matrix with eigenvalue 1, i.e. $\vec{x}T = \vec{x}$. We are interested in the relative fraction $R = x_3^*/x_4^*$, which is the fraction of occupation between the drift-robust and drift-fragile peaks and turns out to be

$$R = \frac{u_{41}\pi_{41}u_{12}\pi_{12}u_{23}\pi_{23}}{u_{14}\pi_{14}u_{21}\pi_{21}u_{32}\pi_{32}}.$$

$$(4)$$

We first calculate the fractions $P_{ij} = \frac{\pi_{ij}}{\pi_{ji}}$. Using Kimura's probability of fixation[44] (a small $s$ approximation of the exact formula of Sella and Hirsh[45, 46]) for an asexual Wright-Fisher process ($N$ is the population size) we find

$$P_{14} = e^{2(s+\epsilon)(N-1)},$$

$$(5)$$

$$P_{12} = P_{23} = e^{s(N-1)},$$

$$(6)$$

so that

$$R = \frac{u_{41}}{u_{14}}\frac{u_{12}}{u_{21}}\frac{u_{23}}{u_{32}}P_{12}P_{23}/P_{14} \equiv M e^{2\epsilon(N-1)},$$

$$(7)$$

where we introduced $M = \frac{u_{41}}{u_{14}}\frac{u_{12}}{u_{21}}\frac{u_{23}}{u_{32}}$, which we now estimate.

For simplicity, we assume that a deleterious mutation rate (for example, the "back mutation rate" $u_{41}$) is given by $\mu$, the overall mutation rate (thus assuming that most mutations are deleterious). We also assume that the mutation rate up the drift-fragile peak ($u_{12} = u_{23} = u_{fragile}$) is greater than the mutation rate up the drift-robust peak ($u_{14} = u_{robust}$); this is equivalent to assuming that small-effect mutations are more frequent than large-effect mutations. Then, one gets

$$M = \frac{u_{41}}{u_{14}}\frac{u_{12}}{u_{21}}\frac{u_{23}}{u_{32}} = \frac{\mu}{u_{robust}}\frac{u_{fragile}}{\mu}\frac{u_{fragile}}{\mu} = \frac{u_{robust}}{\mu}\frac{u_{robust}}{u_{fragile}}$$

$$(8)$$

As $u_{robust} < u_{fragile}$, and $u_{robust} < \mu$ by definition, we find that

$$M < 1,$$

$$(9)$$

thus allowing for a transition between the two peaks determined by the population size $N$.

If we assume beneficial mutations follow a certain distribution, we can derive a precise critical population size at which this transition between fitness peaks occurs. Assume the beneficial mutation rate is $p_b(s) = u_b \mu \rho(s)$, where $\rho(s)$ is the distribution function of mutations with benefit $s$, and $u_b$ is the likelihood that a mutation is beneficial. If mutations with larger benefit $s$ are exponentially more unlikely (see, e.g.,[47–49]), we can use the distribution function $\rho(s) = \frac{1}{\bar{s}} e^{-s/\bar{s}}$ (here $\bar{s}$ is the average beneficial effect) to show that

$$\frac{p_b^2(s/2)}{p_b(s)} = \frac{u_b}{\bar{s}} \ . \tag{10}$$

Then, for $\epsilon$ small we find $\frac{u_{14}}{u_{41}} = p_b(s - \epsilon) \approx p_b(s)$, while $\frac{u_{12}}{u_{21}} = \frac{u_{23}}{u_{32}} = p_b(s/2)$, so that

$$M = p_b(s/2)^2 / p_b(s) = \frac{u_b}{\bar{s}} \equiv \kappa < 1. \tag{11}$$

This result is expected to be general, as it simply states that the flux of beneficial mutations towards the peak with a shallower slope (smaller $s$, here $1 + s/2$) is larger than the flux into the branch with steeper slope (larger $s$, here $1 + s - \epsilon$).

The critical point at which both the drift-fragile and the drift-robust peak are equally populated is determined from setting $R = 1$ in Eq. (7), which gives

$$N_{\text{crit}} = 1 + \frac{\log \kappa^{-1}}{2\epsilon} \ . \tag{12}$$

We show the critical population size as a function of the fitness deficit of the drift robust peak $\epsilon$ in Fig. 3. We see that, depending on the fitness deficit, the evolutionary dynamics prefer the drift-robust peak at small population sizes even though its peak height is inferior to the drift-fragile peak. These results do not depend on the explicit function we used to describe the distribution of beneficial mutations as long as that function is decreasing, nor does it depend on the specifics of the construction of the fitness landscape.

Next, we created an extended version of our fitness-landscape model. Drift-fragility could be exacerbated by subdividing the height $w_3 = 1 + s$ into $n$ increments (Fig. 2b, in the previous model $n = 2$). In this case,

$$\frac{p_b^n(s/n)}{p_b(s)} = \left( \frac{u_b}{\bar{s}} \right)^{n-1}, \tag{13}$$

and the critical population size becomes

$$N_{\text{crit}} = 1 + (n - 1) \frac{\log \kappa^{-1}}{2\epsilon} \ . \tag{14}$$

The simple result Eq. (14) relies on an exact cancellation of the fixation probabilities of the intermediate $n - 1$ steps, and occurs for both the Kimura approximation as well as the exact Sella-Hirsh formula.

**Avida**. Experimental evolution was carried out using the digital evolution system Avida version 2.14. Avida has previously been used to study many concepts that are difficult to test with biological systems[50–55]. In Avida, a population of self-replicating computer programs undergoes Darwinian evolution. Each of the programs ("avidians") consists of a genome of sequential computer instructions, drawn from an alphabet of twenty-six possible instructions. Together, these instructions encode the ability for an avidian to create a new daughter avidian, copy its genome into the new avidian, and divide off the offspring. During this process, mutations can be introduced into the offspring's genome at a controlled rate, introducing genetic variation into the population. When a new offspring is placed into the population (and the population is at carrying capacity), a random individual is replaced by the new avidian, a process that introduces genetic drift into Avida populations. Avidians differ in their replication speed due to different genomic sequences, so avidians that can replicate faster will out-compete slower-replicating types. Therefore, because variation is heritable, and because this variation leads to differential reproduction, an Avida population undergoes Darwinian evolution by natural selection.

The Avida world consists of a toroidal grid of $N$ cells, where $N$ is the maximum population size. Each cell can be occupied by at most one avidian, although a cell may be empty. Upon reproduction, the offspring avidian is placed into an empty cell (if the population is below capacity) or into a random cell, where it replaces the already-present avidian. Although the default Avida setting places offspring into one of nine neighboring cells (including the parent) so as to emulate growth on a surface, in the present experiments any cell may be selected for replacement to simulate a well-mixed environment. Reproduction is asexual in all of the experiments performed here.

Time in Avida is set according to "updates" (the time it takes for an avidian population to execute a give number of instructions). During each update, $30N$

instructions are executed across the population, where $N$ is again the population size. In order to be able to execute its code, an avidian must have a resource, measured as "Single Instruction Processing" units (SIPs). At the beginning of each update, SIPs are distributed to programs in the population in proportion to a quantity called "merit", which is related to a genotype's ability to exploit the environment (see[29] for details). In the experiments performed here, merit was held constant across all individuals, so on average 30 SIPs were distributed to each individual every update.

It should be noted that in most Avida experiments, populations can evolve the ability to perform certain Boolean logic calculations that can improve their merit and hence their fitness[56]. In the experiments performed here, the evolution of these logic calculations was set to be neutral and not under positive selection. Instead, the route for an avidian to improve its fitness was solely by reducing the number of instruction executions needed to copy its genome. A population will typically evolve a faster replication speed by increasing the number of instructions that copy instructions from the parent genome to the offspring genome. When this copy number increase occurs, more instructions are copied per update, resulting in faster replication and greater fitness. This fitness landscape was used because the fitness landscape where logic calculations are under selection lack small-effect deleterious mutations, which would preclude the observation of drift robustness. This lack of small-effect deleterious mutations occurs due to antagonistic pleiotropy and trade-offs between the logic functions and genome replication. Because there are a fixed number of loci in the genome, the more loci dedicated to the logic functions, the fewer loci dedicated to genome replication. Therefore, because most loci are dedicated to logic functions, and mutations to these loci are strongly-deleterious, there are few small-effect deleterious mutations in the logic-function fitness landscape.

Although Avida uses the update as its unit of time, experiments such as those performed here are often run for a given number of generations (the time it takes for the entire population to be replaced). The experiment ends when the average generation across all of the individuals in the population reaches a pre-specified number. Each individual's generation counter is equal to its parent's generation plus one. Therefore, while Avida experiments occur for a set number of generations, the population does not evolve with discrete generations. If fitness differs between individuals and lineages in the population, there can be variation in the individuals' generations in the population.

**Experimental design**. We performed three sets of experiments here. First, initial adaptation experiments were performed to generate genotypes adapted to small and large population-size environments. We evolved 100 small populations ($10^2$ individuals) and 100 large populations ($10^4$ individuals) for $10^5$ generations. The genomic mutation rate was set to $10^{-1}$ mutations/generation/genome and these mutations occurred upon division; offspring could differ by at most one mutation from their parent. The ancestor organism for the initial adaptation treatments was the default Avida ancestor, but with an altered genome length of 50 instructions. This alteration was performed by removing 50 nop-C instructions from the default genome (these instructions are inert).

The second experimental step was to perform a test to measure the drift robustness of individuals evolved at a small population size vs. individuals evolved at a large population size. From each small and large population, we used the most abundant individual to form a set of 100 small-population genotypes and 100 large-population genotypes per treatment. For each of these genotypes, we evolved 10 populations (2000 replicates in total) at a population size of 50 individuals for $10^3$ generations. All other treatment parameters were the same as the initial adaptation experiments.

The final set of experiments tested whether deleterious mutations were responsible for the evolution of drift robustness in small populations. We repeated the initial adaptation experiment and the drift robustness test under the same parameter settings as for the original treatment. However, during the initial adaptation experiment, we reverted any deleterious mutations that appeared in the population[57]. In this setup, the Avida world examines the fitness cost of every new point mutation. If this new mutant has decreased fitness relative to its parent, the mutant is prevented from entering into the population.

**Data analysis**. We calculated statistics for the evolved avidians using Avida's Analyze Mode[29]. In Analyze Mode, the experimenter can run an avidian through its life-cycle (until reproduction) and calculate several genotype characteristics. Fitness was calculated as the ratio between the number of instructions in the genome (the sequence length) to the number of instruction executions needed to copy the genome and reproduce (this is an unbiased predictor of the actual number of offspring).

In order to calculate the distribution of fitness effects for each genotype and other related mutational measures, each point mutation was generated for each genotype ($25 \times L$ mutations, where $L$ is the number of instructions in the genome). The fitness effect of each mutation was calculated as $s = \frac{w_m}{w_0} - 1$, where $w_m$ is the fitness of the mutant and $w_0$ was the fitness of the genotype. The average mutational effect of each genotype is the arithmetic mean of these fitness effects. The fraction of mutations of a given fitness effect was calculated as the number of mutations with that fitness effect divided by $25L$.

To estimate the distribution of fitness effects of fixed mutations for each genotype, we analyzed the line-of-descent (LOD) of these genotypes. The LOD of a genotype contains every genotype that led from the ancestral genotype to the genotype of interest; it represents a fossil record of that lineage[56]. We calculated the fitness effect of each mutation along the LOD as above. For calculating the change in the frequency of lethal and deleterious mutations along the LOD as in (Fig. 6c), we performed the calculations detailed above for each LOD genotype.

In order to examine possible differences in the distribution of beneficial fixed effects between small populations and large populations, we had to identify the beneficial mutations that contributed to adaptation. This is non-trivial, as small populations fix more beneficial mutations than large populations due to their oscillations in fitness. In order to not include these transient fixed beneficial mutations, we selected the beneficial mutations from each population whose fitness gain was at least partially maintained during the future evolution of the population. We labeled a beneficial mutation on a population's LOD as maintained if 1) it resulted in the lineage attaining a new fitness maximum, and 2) fitness never decreased below the previous fitness value on the LOD except for a transient amount of time. We defined a transient amount of time as less than five consecutive genotypes on the LOD having a lower fitness. This transient fitness decrease allowance is necessary due to the possibility of valley-crossing in Avida fitness landscapes[57].

To compare the fraction of small-effect deleterious mutations between genotypes from small populations and genotypes from large populations (Fig. 7), we first selected one genotype from each lineage for a given fitness value. If a lineage had multiple genotypes with the same fitness, as was often the case, we took the last genotype that appeared. Then, for each fitness value with more than 20 genotypes from both small and large populations, we calculated the fitness effect of every possible point mutation and the fraction of these mutations that were deleterious with a small effect size as described above.

Statistical analyses were performed using the NumPy[58], SciPy[59], and Pandas[60] Python modules. Figures were created with the Matplotlib[61] Python module. The stationary distribution for the mathematical model was solved using Mathematica version 11.0.1.0[62].

**Data availability**. The Avida software is available for free use (https://github.com/devosoft/avida). Avida configuration scripts, data from Avida experiments, statistical analysis and figure-generating scripts, as well as the Mathematica code, are available at the Dryad data repository (DOI:10.5061/dryad.nr780).

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

## Acknowledgements

T.L. acknowledges a Michigan State University Distinguished Fellowship, a BEACON fellowship, and the Russell B. Duvall award for support. This work was supported in part by Michigan State University through computational resources provided by the Institute for Cyber-Enabled Research. This material is based in part upon work supported by the National Science Foundation under Cooperative Agreement No. DBI-0939454. Any opinions, findings, and conclusions or recommendations expressed in this material are those of the authors and do not necessarily reflect the views of the National Science Foundation.

## Author contributions

TL performed the experiments and data analysis. TL and CA designed the experiments and wrote the paper.

## Additional information

**Competing interests:** The authors declare no competing financial interests.

