## [Peer Review File · Nature Communications]

Reviewers' comments:

Reviewer #1 (Remarks to the Author):

This manuscript is grounded in a result that is simple to understand in retrospect, but that I do not believe has been pointed out before. The manuscript is about “drift robustness”, defined as the susceptibility of a population to future declines in fitness due to the fixation of slightly deleterious mutants. The central result is a simple consequence of conditional probability: given that a population is observed at a random time during its evolutionary trajectory through genotype space, the population is unlikely to reside in a location that it has a high propensity to leave. Because small populations have a high propensity to leave genotypes subject to many slightly deleterious mutations, small populations are therefore unlikely to be observed with such genotypes. The same is not true for large populations, and so small populations have higher drift robustness than large populations. However, the high drift robustness of small populations is not an adaptation against drift because of their past history experiencing drift and adaptation to its consequences, but instead is a non-adaptive consequence of the workings of conditional probability. Consequences of conditional probability are notoriously hard to spot, and this manuscript makes a useful contribution by spotting this one.

However, the current organization of the manuscript obscures the underlying simple nature of the result, seriously reducing the interest of the manuscript to non-specialists, or indeed, even to specialists like myself. I suspect that this might be because the manuscript at least partially follows the historical order in which results were obtained – an oddity was observed using Avida simulations, then extensive testing was done in the Avida system, then the underlying nature of the result became clear, and finally its generality is argued for using a much simpler model. I rarely say this, but the extent of the testing in this paper far exceeds what is needed. This is because that once the nature of the central result is understood, it is so compelling that little further persuasion is required. As a result, reading through all the rigorous testing is tedious, and delays the presentation of the result in greater generality. If I weren't reviewing the paper, I would have had a hard time reading to the end.

I strongly recommend that the manuscript be entirely reorganized and in parts rewritten according to a different scheme, following at least approximately the following outline. First, the general nature of the result should be argued for verbally a priori, rather than being introduced later as the explanation for an observation. Second, the simple model illustrated in Supplementary Figure 10 should be used to make the idea formal and concrete. Third, a discussion of the generality of this model and how it applies to more complex genotype spaces should follow, given interest in origin-fixation models of this kind, eg by David McCandlish and others. Fourth, Avida results should be presented, not as currently for the purpose of proving that the result exists, but instead for demonstrating the magnitude of the effect in a semi-realistic genotype space. The suddenness of the effect described in lines 242-254 is also of interest at this point, as part of the “mode” as well as the magnitude of the effect. Avida results may also be useful to show how this magnitude persists with relaxation of strict origin-fixation dynamics. The emphasis of Avida results should be completely transformed from proof of what drives the result to an exploration of its

magnitude.

As part of this, I would like to see greater integration of the main point with previous literature, eg Krakauer & Plotkin ref 9 (see the Introduction of <https://doi.org/10.1534/genetics.116.192567> for our own commentary on ref 9). Eg, it seems to me that empirical or even theoretical findings that might be seen to support the cost assumptions of ref 9 might in fact be explained by the alternative phenomenon described in this manuscript. This would benefit from a detailed and engaged discussion of the interplay between alternative viewpoints, rather than simple "they say this" citations eg as currently found on lines 332-337.

Minor points and typos:

- The Abstract and Introduction refer to "drift load", but my understanding of the term "drift load" is different, and included, albeit not yet under that name, in Kimura 1963 ref 3 in what the Introduction calls "mutation load", as the special portion of mutation load that is higher than usual in small populations. "Load", whether mutation load, drift load, substitutional load, segregational load etc., is defined as the reduction in population mean fitness relative to some standard, with the standard normally being either the globally maximum fitness or the fittest individual within the population. But this is not how the term is used in this manuscript; drift load seems indistinguishable from drift robustness. I realize that there are exceptions to this, eg Whitlock 2000 ref 19, but they are exceptions not the rule and can potentially be justified using the global mean fitness standard that is not accessible in Avida and not used in this manuscript. For clarity with a larger body of work on load, I therefore recommend the elimination of the term "load", and the exclusive use of the term drift robustness throughout.

- line 33: shown -> proposed: the cited refs are theory-based not empirical. Ref 19 should be added to the cited refs for this line.

- line 65: lower case for Drift

- line 404 were -> where

- line 527 Fischer -> Fisher

- line 545 prefers -> prefer

- Code availability: I would also like to see the statistical analysis codes and Mathematica notebook released on github.

- Figure 2: Legend says "Colors as in Figure 1" but they are not simply black and red. Difficulty parsing the colors makes the figure very hard work for the reader.

- Supplementary Figure 7: Legend is incomplete.

Signed,

Joanna Masel

Reviewer #2 (Remarks to the Author):

The theme of the paper is interesting and the methodology used to address it very appropriate. The paper introduces an important new concept, one that will be relevant to a range of researchers in evolutionary biology. Avida is a great system to ask and answer these questions, and the authors have done much work to build and support their argument. I very much appreciated the addition of the theoretical model, it nicely complements the Avida work. Overall I find the paper generally appropriate for the journal, but still rough around the edges and in need of some medium-level editing.

The majority of my concerns could be categorized in three categories:

- overstating the conclusions, going beyond what the data is showing
- conclusions entirely not supported by data presented
- things poorly explained/confusing, not well labeled, etc.

The comments are detailed point by point below, more or less following the order of the paper.

Even though it is quite visually convincing, some stats would be nice to support the conclusion about the existence of drift robustness (Figure 1)

For Figure 2, it would be nice to explain the different colors (pink, red, gray) and their meaning and include a legend in one of the panels.

Legend for the Figure 7 is incomplete (no explanation of the panels b and c). The b and c panels seem to have the same axis, unclear what the difference is. Also, what is there is difficult to read, for example, I stumbled over "the ratio of the fraction" in the "a) a function of the ratio of the fraction of small-effect deleterious mutations in the small-population genotype to the fraction of small-effect deleterious mutations in the large-population genotype," I would suggest including a formula to make this more clear.

Line 144- 168: I really dislike the "correlation hunting" that seems to be going on in this section. A mechanistic explanation, even one that is a hypothesis, would be much preferred. Like this, it remains quite unconvincing.

Line 139: Why use only 3 small but 12 large populations in some of the experiments? A more balanced experimental design would make more sense.

Sup Figure 8 is entirely uninformative. Use a log-y scale, zoom in, do something, but one

cannot see anything in this presentation. There is no point in showing it like this.

The scale in Sup Figure 9 is peculiar and made me wonder. Do the individuals from small and large populations have *exactly* the same fitness? Or are there still differences after some decimal places. After eventually reading that there were no tasks were rewarded this seems more likely, as all that would be needed is exactly the same gestation time for the pairs. Is this indeed the case?

Line 43: Strictly speaking, avidians do not "compete for the resources necessary for reproduction". They compete for space, the resources for reproduction are unlimited, at least in the standard Avida setup. Please clarify.

Line 56: Drift should not be capitalized.

Line 70: Fitness trajectories for small and large populations are mentioned, but only the ones for large populations are shown in the SI

Line 104: "We 104 confirmed that these trends hold when calculated per genotype (rather than averaging over 105 genotypes) as well." This is unclear to me. Please explain better what you mean, which graph this refers to, which of the stats in the text.

Line 117: Correlation is not causation!!! "These two variables are strongly anti-correlated (Pearson's $r = -0.92$; $117 p < 4 \times 10^{-84}$), demonstrating that a genotype's drift robustness is determined by the fraction 118 of small-effect mutational neighbors (Fig. 3b)." Even with the relatively ambiguous verb, "determines", this conclusion just does not stand, especially since you set up the test using "cause". This is surprisingly sloppy.

Line 161: The Supplementary Table 1 does not support the text. Based on the numbers noted, none of the correlations calculated for individual genotypes are significant.

Line 222: I don't buy this argument. Yes, the symmetrical distribution is consistent with oscillations, but the asymmetrical one point to oscillations as well, just not with equal number of beneficial and deleterious mutations. Given the power that you have with Avida, and given that you already have the lineages, there is no reason to not examine this directly, over time.

Line 224: I'm often having small but significant issues with the phrasing, which implies more than you show. "The previous two tests demonstrate that small populations cannot climb drift-fragile 224 fitness peaks." Actually, they show that they do not, in your experiments. Whether or not they can, under some conditions, we have no idea.

Line 281: "...large populations could also adapt to drift-robust peaks if 277 they had evolved for as many generations as the small populations. We present two pieces of 278 evidence that suggest that this is extremely unlikely." The authors go on to argue, in their first point, that the populations have reaches *a* fitness peak. Although that is true, they provide no evidence that they will not move from that peak to a more drift-robust one later

in evolution.

Line 281: I'm sorry, maybe I'm missing something, but I just don't get the second part of this argument. I simply don't see how the equal fitness or the difference in proportion of small-effect deleterious mutations implies anything about the effect of the length of evolution time that is needed to potentially reach a drift-robust peak. I am not saying that I think the large populations will do it, just that the evidence and arguments presented are insufficient to claim it will not.

Line 317: "The results for this model confirm that the evolution of drift robustness in small 317 populations is a general phenomenon and not an Avida-specific result." Again, I feel this is slightly overstated. It certainly implies that it is not just specific to Avida, but is more general and applies to some other systems. Saying it's general implies it is present everywhere, for which there is no support. Authors are much more careful later, where they nicely outline some of the requirements for drift-robustness to appear.

Line 336: "Neither of these characteristics were found in our experiments." Authors should discuss why they have not been found in their experiments.

Line 402: The discussion of merit is confusing, for someone who knows a bit about Avida, and especially due to citing [29] which also describes merit in terms of tasks, etc. I would bring up the fact that they are no logic tasks being rewarded a bit sooner. Also, I would like to hear an argument as to why not, since this is a typical and integral part of evolution in Avida.

Sup Figure 10: The legend should be expanded and the figure properly explained.

Reviewer #3 (Remarks to the Author):

This is an excellent paper that will be widely influential, I think.

I have mainly "minor" suggestions, but some of these are critical to making this paper easier to read. In particular the figures and the figure captions can be greatly improved.

One issue that I think substantially takes away from the clarity of the paper is that the authors chose to keep total number of mutations constant rather than total evolutionary time (generations). Of course they tested the effect of this scaling in the supplemental figures, but this is a strange choice that makes the reader constantly wonder if this is the root cause of various features (e.g., see next paragraph). Nature does not keep number of mutations constant between large and small populations, but they could be fairly judged after similar periods of time had past. I would strongly recommend using results derived from comparable evolutionary time in all results presented in the main text of the paper. Lines 279-281, for example, argue that the large-populations have reached fitness plateaus, but that is not obvious to me from the figure and would be much better addressed by

simply having longer runs for the large populations to compare.

Supplemental figure 1 is confusing. It seems to show that the small populations have higher mean fitness than the large populations. This seems to escape comment in the main paper. I presume this is due to the fact that the small populations have had more time for evolution to work, but I don't know. If it is due to the difference in evolutionary time, it really reinforces the point in my previous paragraph.

The aspect of this paper that I find most confusing and unsatisfying is in the section that starts on line 133. No satisfactory explanation is given for why the small population genotypes outcompete the large-population genotypes, especially at large population size. These genotypes were apparently chosen to have the same growth rates, so it is confusing why this should be. It might be fruitful to run these experiments again with mutation set to 0 to confirm that the fitness is actually equal in the absence of new mutations (and therefore removing robustness from the assay).

I really liked the section that starts at line 256.

The bar graphs in most of the figures would be much easier to read if the two cases were presented with the bars side by side rather than overlapping.

33: Whitlock 2000 in Evolution showed that beneficial mutations could halt drift load.

65: "Drift" should probably not be capitalized

405: "were" → "where"

585: Confusing: both mutations were non-viable but also not lethal?

Figure 2 caption: "Colors as in figure 1" This isn't actually true. Figure 1 has black for small populations and figure 2 has grey.

Figure 2: Move the description of boxplots to the description of 2d.

Figure 3: The axis labels need improved clarity. "Fraction of mutations" is not clear without recourse to the caption. "Fraction of weakly deleterious mutations" or similar would be much clearer.

Figure 5d: I really don't know what "likelihood of different mutational effects" means. Also, lethal is a subset of deleterious, so how can black ever exceed red? Finally, read and black have been used previously to distinguish small and large, so a different color contrast here would be better.

Figure 6 caption: "relative fitness before and after 10^3 generations" is a little confusing – I was looking for two values. "fitness after 10^3 generations relative to before" is maybe

clearer?

Supplemental Table 1: It is not clear what the value of this table is.

Supplemental Figure 3: Again the y-axis labels need better clarity. Panel a and b have the same label, but I think they mean different things on the two panels. This is very confusing to the reader.

Supplemental figure 7: caption does not refer at all to the panel c.

Supplemental figure 9: Bonferroni should be capitalized.

Supplemental figure 11: Axes labels should be labeled.

We thank the reviewers for their comments. We have made extensive changes to the manuscript based on the comments from reviewers 1 & 3; we summarize the main changes here.

1. We repeated our Avida experiments and evolved both small and large populations for an equal number of generations (10^5 generations). All data now presented in this manuscript are from these new experiments, but the results are qualitatively the same as with our previous experiments (where small populations evolved for more generations than large populations).

2. We have restructured the Results section in accordance with reviewer 1's comments. We first propose our (verbal) hypothesis in the Introduction, then start the Results section with our mathematical model. We end by presenting the results from our Avida experiments to test the hypothesis in a complex fitness landscape. Due to the restructuring of the manuscript and the new experiments, we have removed some of our Avida results in order to make the manuscript easier to understand. We have also changed the Methods section to reflect the new order of the results and added or removed content as required.

Below, we have listed the comments given by each reviewer in italics and wrote our responses below each comment in bold font. Our changes to the main text are denoted in blue.

Reviewer #1 (Remarks to the Author):

This manuscript is grounded in a result that is simple to understand in retrospect, but that I do not believe has been pointed out before. The manuscript is about “drift robustness”, defined as the susceptibility of a population to future declines in fitness due to the fixation of slightly deleterious mutants. The central result is a simple consequence of conditional probability: given that a population is observed at a random time during its evolutionary trajectory through genotype space, the population is unlikely to reside in a location that it has a high propensity to leave. Because small populations have a high propensity to leave genotypes subject to many slightly deleterious mutations, small populations are therefore unlikely to be observed with such genotypes. The same is not true for large populations, and so small populations have higher drift robustness than large populations. However, the high drift robustness of small populations is not an adaptation against drift because of their past history experiencing drift and adaptation to its consequences, but instead is a non-adaptive consequence of the workings of conditional probability. Consequences of conditional probability are notoriously hard to spot, and this manuscript makes a useful contribution by spotting this one.

However, the current organization of the manuscript obscures the underlying simple nature of the result, seriously reducing the interest of the manuscript to non-specialists, or indeed, even to specialists like myself. I suspect that this might be because the manuscript at least partially follows the historical order in which results were obtained – an oddity was observed using Avida simulations, then extensive testing was done in the Avida system, then the underlying nature of the result became clear, and finally its generality is argued for using

a much simpler model. I rarely say this, but the extent of the testing in this paper far exceeds what is needed. This is because that once the nature of the central result is understood, it is so compelling that little further persuasion is required. As a result, reading through all the rigorous testing is tedious, and delays the presentation of the result in greater generality. If I weren't reviewing the paper, I would have had a hard time reading to the end.

I strongly recommend that the manuscript be entirely reorganized and in parts rewritten according to a different scheme, following at least approximately the following outline. First, the general nature of the result should be argued for verbally a priori, rather than being introduced later as the explanation for an observation. Second, the simple model illustrated in Supplementary Figure 10 should be used to make the idea formal and concrete. Third, a discussion of the generality of this model and how it applies to more complex genotype spaces should follow, given interest in origin-fixation models of this kind, eg by David McCandlish and others. Fourth, Avida results should be presented, not as currently for the purpose of proving that the result exists, but instead for demonstrating the magnitude of the effect in a semi-realistic genotype space. The suddenness of the effect described in lines 242-254 is also of interest at this point, as part of the "mode" as well as the magnitude of the effect. Avida results may also be useful to show how this magnitude persists with relaxation of strict origin-fixation dynamics. The emphasis of Avida results should be completely transformed from proof of what drives the result to an exploration of its magnitude.

As per this recommendation, we have re-written the manuscript broadly in accordance with the above outline. We present our hypothesis in verbal form, then our mathematical model, and finally the Avida results. We have removed some of the previous Avida analyses used to test the basis of drift robustness in Avida, although we still dedicate the majority of the manuscript to the Avida results. We have included more results of the sudden transition to drift robustness (Figure 7b-d) and results exploring how many populations fix beneficial mutations that decrease the likelihood of deleterious mutations.

As part of this, I would like to see greater integration of the main point with previous literature, eg Krakauer & Plotkin ref 9 (see the Introduction of <https://doi.org/10.1534/genetics.116.192567> for our own commentary on ref 9). Eg, it seems to me that empirical or even theoretical findings that might be seen to support the cost assumptions of ref 9 might in fact be explained by the alternative phenomenon described in this manuscript. This would benefit from a detailed and engaged discussion of the interplay between alternative viewpoints, rather than simple "they say this" citations eg as currently found on lines 332-337.

We have included two paragraphs discussing how our results compare to the existing literature:

We are not the first to propose that small populations will evolve robustness mechanisms in response to their deleterious mutational burden. However, these mechanisms are usually discussed in terms of mutational robustness, not robustness to drift. Previous studies provided two characteristics of the evo-

lution of mutational robustness in small populations. First, small populations should preferentially evolve to lower fitness peaks with more “redundancy,” defined as a decreased average deleterious mutational effect and large populations should evolve to fitness peaks with a high average deleterious mutational effect (Krakauer and Plotkin 2002, Elena et al. 2007). Our results suggest opposite evolutionary trajectories for small and large populations. While our results concerning the evolution of drift robustness do suggest that small populations evolve to lower fitness peaks, and small populations do evolve more redundancy in terms of exactly-neutral mutations, these small populations do not evolve towards fitness peaks with a decreased deleterious mutational effect (Figure 4d). In fact, they evolve towards fitness peaks with a minimal likelihood of small-effect deleterious mutations. This discrepancy likely exists due to the fitness landscape used to study the evolution of redundancy in small populations: the mutations in that fitness landscape were all small-effect deleterious mutations (Krakauer and Plotkin 2002). Thus, small populations could not maintain fitness except on the flattest of fitness landscapes (Krakauer and Plotkin 2002). In a version of this model with multiple fitness peaks (e.g., one with small-effect deleterious mutations and one with large-effect deleterious mutations), we expect that small populations would evolve to the peak with large-effect deleterious mutations. Such an outcome was recently predicted for populations evolving at very high mutation rates (Lan et al. 2017), although a different model predicts that small populations should evolve to areas that minimize the deleterious effect of mutations (Gros et al. 2009), in accordance with Krakauer and Plotkin’s model (Krakauer and Plotkin 2002).

The second characteristic of mutational robustness in small populations is that these populations should evolve “global” robustness mechanisms, such as error-correction mechanisms, that affect many loci (Gros et al. 2009, Rajon and Masel 2011, Xiong et al. 2017). There are no global error-correction mechanisms available to the *Avidia* genomes here (although one could allow the evolution of mutation rates, e.g., Clune et al. 2008). However, we did find that small populations preferentially fixed epistatic mutations that strongly altered the likelihood of deleterious mutations (Figure 7c,d). These mutations are global in the sense that they alter the fitness effects of mutations at multiple loci. However, unlike previous work that suggested small populations should fix global solutions that reduce the effect of deleterious mutations (Gros et al. 2009, Rajon and Masel 2011, Xiong et al. 2017), we found that these mutations increased the likelihood of lethal mutations. We do not have strong evidence that this increased lethality is essential and expect that small populations could also fix mutations that increased the likelihood of neutral mutations while reducing the likelihood of deleterious mutations if they exist in the *Avida* fitness landscape. Generally, our results emphasize that the evolutionary process behind drift robustness is the trend to reduce the likelihood of small-effect deleterious mutations, which can be achieved in multiple ways.

Minor points and typos:

- The Abstract and Introduction refer to “drift load”, but my understanding of the term “drift load” is different, and included, albeit not yet under that name, in Kimura 1963 ref 3 in what the Introduction calls “mutation load”, as the special portion of mutation load that is higher than usual in small populations. “Load”, whether mutation load, drift load, substitutional load, segregational load etc., is defined as the reduction in population mean fitness relative to some standard, with the standard normally being either the globally maximum fitness or the fittest individual within the population. But this is not how the term is used in this manuscript; drift load seems indistinguishable from drift robustness. I realize that there are exceptions to this, eg Whitlock 2000 ref 19, but they are exceptions not the rule and can potentially be justified using the global mean fitness standard that is not accessible in *Avida* and not used in this manuscript. For clarity with a larger body of work on load, I therefore recommend the elimination of the term “load”, and the exclusive use of the term drift robustness throughout.

We used the term “drift load” to refer the reduction in fitness due to the fixation of slightly-deleterious mutations in small populations. Given that the evolution of drift robustness also results in a reduced fitness for small populations, we can see how our use of the term may appear unclear. We have removed all instances of drift load from the manuscript.

- line 33: shown -> proposed: the cited refs are theory-based not empirical. Ref 19 should be added to the cited refs for this line.

- line 65: lower case for Drift

- line 404 were - > where

- line 527 Fischer - > Fisher

- line 545 prefers - > prefer

We have made the above changes where still applicable

- Code availability: I would also like to see the statistical analysis codes and Mathematica notebook released on github.

We will include the statistical analyses code and the mathematica notebook in the Dryad depository.

- Figure 2: Legend says “Colors as in Figure 1” but they are not simply black and red. Difficulty parsing the colors makes the figure very hard work for the reader.

- Supplementary Figure 7: Legend is incomplete.

We have improved our figures (i.e., no overlapping bar graphs) and specified the difference between black and gray where necessary.

Signed,

Joanna Masel

Reviewer #2 (Remarks to the Author):

The theme of the paper is interesting and the methodology used to address it very appropriate. The paper introduces an important new concept, one that will be relevant to a range of researchers in evolutionary biology. Avida is a great system to ask and answer these questions, and the authors have done much work to build and support their argument. I very much appreciated the addition of the theoretical model, it nicely complements the Avida work. Overall I find the paper generally appropriate for the journal, but still rough around the edges and in need of some medium-level editing.

The majority of my concerns could be categorized in three categories:

- overstating the conclusions, going beyond what the data is showing*
- conclusions entirely not supported by data presented*
- things poorly explained/confusing, not well labeled, etc.*

The comments are detailed point by point below, more or less following the order of the paper.

Even though it is quite visually convincing, some stats would be nice to support the conclusion about the existence of drift robustness (Figure 1)

We have included statistics on the presence of drift robustness in the results from Figure 5a and Figure 8c.

For Figure 2, it would be nice to explain the different colors (pink, red, gray) and their meaning and include a legend in one of the panels.

In our new figures, we have removed the overlapping bars of different colors and clearly stated what each color means.

Legend for the Figure 7 is incomplete (no explanation of the panels b and c). The b and c panels seem to have the same axis, unclear what the difference is. Also, what is there is difficult to read, for example, I stumbled over "the ratio of the fraction" in the "a) a function of the ratio of the fraction of small-effect deleterious mutations in the small-population genotype to the fraction of small-effect deleterious mutations in the large-population genotype," I would suggest including a formula to make this more clear.

Line 144-168: I really dislike the "correlation hunting" that seems to be going on in this section. A mechanistic explanation, even one that is a hypothesis, would be much preferred. Like this, it remains quite unconvincing.

Line 139: Why use only 3 small but 12 large populations in some of the experiments? A more balanced experimental design would make more sense.

Here is what we think occurred in the competition experiments. There did not seem to be a relationship between a decreased likelihood of small-effect deleterious mutations and the number of competitions won by the drift-robust genotype, as we presented. There was a relationship between greater likelihood of *neutral* mutations and drift-robust genotype success. There is some previous justification for this trend, as previous theoretical work (e.g., Van Nimwegen et al. 1999) has shown that there can be selection for neutrality in neutral-network fitness landscapes (i.e., no difference in fitness between genotypes, but only in the likelihood of neutral mutations). We first thought that this could explain our results: selection for neutrality drives the evolution of drift robustness, as genotypes with greater neutrality tend to have a decreased likelihood of small-effect deleterious mutations.

However, when we performed the same competition experiments at our large population size, we saw the same outcome: drift-robust genotypes out-competed drift-fragile genotypes and the probability of the drift-robust genotype sweeping the population was correlated with the fraction of neutral mutations. Thus, we concluded that this could not explain the differences between small populations and large populations as the same dynamics occurs at both population sizes.

In the current version of the manuscript, we have removed the competition experiments entirely.

Van Nimwegen, Erik, James P. Crutchfield, and Martijn Huynen. "Neutral evolution of mutational robustness." *Proceedings of the National Academy of Sciences* 96.17 (1999): 9716-9720.

Sup Figure 8 is entirely uninformative. Use a log-y scale, zoom in, do something, but one cannot see anything in this presentation. There is no point in showing it like this.

This figure has been removed from the manuscript.

*The scale in Sup Figure 9 is peculiar and made me wonder. Do the individuals from small and large populations have *exactly* the same fitness? Or are there still differences after some decimal places. After eventually reading that there were no tasks were rewarded this seems more likely, as all that would be needed is exactly the same gestation time for the pairs. Is this indeed the case?*

Yes, these are genotypes with exactly the same fitness (we do round the fitness values we show in the figure for visualization's sake, but that happens after we group the genotypes).

Line 43: Strictly speaking, avidians do not "compete for the resources necessary for reproduction". They compete for space, the resources for reproduction are unlimited, at least in the standard Avida setup. Please clarify.

We have replaced that sentence (now located in the Results) with the following:

In Avida, a population of self-replicating computer programs ("avidians") compete for the memory space and CPU time necessary for reproduction.

Line 56: Drift should not be capitalized.

Line 70: Fitness trajectories for small and large populations are mentioned, but only the ones for large populations are shown in the SI

The subtitle with "Drift" has been renamed and the fitness trajectory figure has been removed.

Line 104: "We confirmed that these trends hold when calculated per genotype (rather than averaging over 105 genotypes) as well." This is unclear to me. Please explain better what you mean, which graph this refers to, which of the stats in the text.

We have replaced the above sentence with the following:

We confirmed that these trends hold when we calculated a DFE for each genotype (rather than one DFE for all genotypes from a given population size) as follows.

We have identified the relevant figure, if one is in the manuscript, after each calculated statistic.

Line 117: Correlation is not causation!!! "These two variables are strongly anti-correlated (Pearson's $r = -0.92$; $p < 4 \times 10^{-84}$), demonstrating that a genotype's drift robustness is determined by the fraction of small-effect mutational neighbors (Fig. 3b)." Even with the relatively ambiguous verb, "determines", this conclusion just does not stand, especially since you set up the test using "cause". This is surprisingly sloppy.

We have changed the text concerning this result to the following:

Furthermore, a genotype's decline in fitness is correlated with its likelihood of a small-effect deleterious mutation, supporting the idea that small populations have evolved to fitness peaks with a low likelihood of small-effect deleterious mutations due to the peak's drift robustness (Spearman's $\rho = 0.80$, $p \approx 0$; Fig. 5b)

Line 161: The Supplementary Table 1 does not support the text. Based on the numbers noted, none of the correlations calculated for individual genotypes are significant.

We have removed the above supplementary table as the competition experiments have been removed.

Line 222: I don't buy this argument. Yes, the symmetrical distribution is consistent with oscillations, but the asymmetrical one point to oscillations as well, just not with equal number of beneficial and deleterious mutations. Given the power that you have with Avida, and given that you already have the lineages, there is no reason to not examine this directly, over time.

Due to the reorganization of this manuscript, we have removed this figure. But we agree, there are better ways of demonstrating this point.

Line 224: I'm often having small but significant issues with the phrasing, which implies more than you show. "The previous two tests demonstrate that small populations cannot climb drift-fragile fitness peaks." Actually, they show that they do not, in your experiments. Whether or not they can, under some conditions, we have no idea.

We have removed the above sentence from the manuscript.

*Line 281: "...large populations could also adapt to drift-robust peaks if they had evolved for as many generations as the small populations. We present two pieces of evidence that suggest that this is extremely unlikely." The authors go on to argue, in their first point, that the populations have reaches *a* fitness peak. Although that is true, they provide no evidence that they will not move from that peak to a more drift-robust one later in evolution.*

Line 281: I'm sorry, maybe I'm missing something, but I just don't get the second part of this argument. I simply don't see how the equal fitness or the difference in proportion of small-effect deleterious mutations implies anything about the effect of the length of evolution time that is needed to potentially reach a drift-robust peak. I am not saying that I think the large populations will do it, just that the evidence and arguments presented are insufficient to claim it will not.

We have removed these and related statements as our new results were generated after evolving both small and large populations for the same number of generations.

Line 317: "The results for this model confirm that the evolution of drift robustness in small populations is a general phenomenon and not an Avida-specific result." Again, I feel this is slightly overstated. It certainly implies that it is not just specific to Avida, but is more general and applies to some other systems. Saying it's general implies it is present everywhere, for which there is no support. Authors are much more careful later, where they nicely outline some of the requirements for drift-robustness to appear.

We have removed this sentence from the manuscript, as the mathematical model is now presented before the Avida results. The paragraph outlining the requirements for drift robustness still remains in the Discussion.

Line 336: "Neither of these characteristics were found in our experiments." Authors should discuss why they have not been found in their experiments.

We have expanded our discussion of these two characteristics in reference to previous literature on the subject (see our response to reviewer 1).

Line 402: The discussion of merit is confusing, for someone who knows a bit about Avida, and especially due to citing [29] which also describes merit in terms of tasks, etc. I would bring up the fact that they are no logic tasks being rewarded a bit sooner. Also, I would like to hear an argument as to why not, since this is a typical and integral part of evolution in Avida.

We have added the following two sentences to the Methods section describing merit:

This fitness landscape was used because the fitness landscape where logic calculations are under selection lack small-effect deleterious mutations, which would preclude the observation of drift robustness.

Sup Figure 10: The legend should be expanded and the figure properly explained.

The former Sup Figure 10 is now Figure 2. We have expanded our description in the caption to the following:

The fitness landscapes for the Markov model to test for the evolution of drift robustness. Each circle represents one genotype and is labeled with its fitness. Each arrow represents the transition between one genotype to another (including the identical genotype) and is labeled with the transition probability. a) The fitness landscape for the minimal model. s represents the selection coefficient of the drift-fragile peak and ϵ represents the small fitness difference between the drift-fragile peak and the drift-robust peak. u_{ij} and π_{ij} represent the mutation rate between genotypes and probability of fixation from one genotype to another, respectively. b) The fitness landscape for the extended model. Variables as in panel a. Transition probabilities omitted for clarity.

Reviewer #3 (Remarks to the Author):

This is an excellent paper that will be widely influential, I think.

I have mainly "minor" suggestions, but some of these are critical to making this paper easier to read. In particular the figures and the figure captions can be greatly improved.

One issue that I think substantially takes away from the clarity of the paper is that the authors chose to keep total number of mutations constant rather than total evolutionary time (generations). Of course they tested the effect of this scaling in the supplemental figures, but this is a strange choice that makes the reader constantly wonder if this is the root cause of various features (e.g., see next paragraph). Nature does not keep number of mutations constant between large and small populations, but they could be fairly judged after similar periods of time had past. I would strongly recommend using results derived from comparable evolutionary time in all results presented in the main text of the paper. Lines 279-281, for example, argue that the large-populations have reached fitness plateaus, but that is not obvious to me from the figure and would be much better addressed by simply having longer runs for the large populations to compare.

Supplemental figure 1 is confusing. It seems to show that the small populations have higher mean fitness than the large populations. This seems to escape comment in the main paper. I presume this is due to the fact that the small populations have had more time for evolution to work, but I don't know. If it is due to the difference in evolutionary time, it really reinforces the point in my previous paragraph.

We repeated our Avida experiments and ran both small and large populations for 10^5 generations. All of the Avida results in the manuscript are now from these experiments and we think they clear up any confusion that would result from evolving the populations for different numbers of generations.

The aspect of this paper that I find most confusing and unsatisfying is in the section that starts on line 133. No satisfactory explanation is given for why the small population genotypes outcompete the large-population genotypes, especially at large population size. These genotypes were apparently chosen to have the same growth rates, so it is confusing why this should be. It might be fruitful to run these experiments again with mutation set to 0 to confirm that the fitness is actually equal in the absence of new mutations (and therefore removing robustness from the assay).

We have removed the competition experiments from the manuscript, as they seem to cause more confusion than clarity. Here is what we think happened (see our comment to reviewer 2 for additional clarification). While there did not appear to be a relationship between the relative lack of small-effect deleterious mutations and competition outcome, there was a relationship between the relative abundance of neutral mutations and competition outcome. As the small-population genotypes had a greater likelihood of neutral mutations, they were out-competing the drift-fragile genotypes. However, this out-competition occurred in both small and large populations, and thus could not explain the disparity between small populations and large populations that we saw in our main experiments.

We did perform these experiments at the no-mutation rate limit as a control and got neutral results, as expected, although these were not included in the

manuscript.

I really liked the section that starts at line 256.

The bar graphs in most of the figures would be much easier to read if the two cases were presented with the bars side by side rather than overlapping.

We have changed all of our bar graph figures so that bars are now side-to-side as opposed to overlapping.

33: Whitlock 2000 in Evolution showed that beneficial mutations could halt drift load.

We have included this citation after the relevant sentence.

65: “Drift” should probably not be capitalized

405: “were” -> “where”

We have fixed both of these grammatical errors.

585: Confusing: both mutations were non-viable but also not lethal?

We have removed this section of the Methods due to removing this analysis from the Results section.

Figure 2 caption: “Colors as in figure 1” This isn’t actually true. Figure 1 has black for small populations and figure 2 has grey.

We have changed the colors in this figure and all other figures to gray and red for consistency

Figure 2: Move the description of boxplots to the description of 2d.

We have moved the description of the boxplots to 4c, where they now first appear.

Figure 3: The axis labels need improved clarity. “Fraction of mutations” is not clear without recourse to the caption. “Fraction of weakly deleterious mutations” or similar would be much clearer.

We have changed the axes titles to “Fraction Small-effect Deleterious Mutations in the current Fig. 4c”

Figure 5d: I really don’t know what “likelihood of different mutational effects” means. Also, lethal is a subset of deleterious, so how can black ever exceed red? Finally, read and

black have been used previously to distinguish small and large, so a different color contrast here would be better.

We changed the figure legend to say : "Likelihood of Mutational Effect" and included a legend in the figure for clarification. Here, and throughout the manuscript, we distinguished lethal mutations from deleterious mutations and define deleterious mutations as non-lethal, fitness-decreasing, mutations. We changed the colors of the lines in Figure 7c to green and magenta.

Figure 6 caption: "relative fitness before and after 10^3 generations" is a little confusing – I was looking for two values. "fitness after 10^3 generations relative to before" is maybe clearer?

We have changed the caption (now in Fig. 8c) to read :

Relative fitness of the most-abundant genotype from every population during the drift robustness test.

Supplemental Table 1: It is not clear what the value of this table is.

Supplemental Figure 3: Again the y-axis labels need better clarity. Panel a and b have the same label, but I think they mean different things on the two panels. This is very confusing to the reader.

Supplemental figure 7: caption does not refer at all to the panel c.

These figures have been removed in the current version of the manuscript.

Supplemental figure 9: Bonferroni should be capitalized.

We have made this correction.

Supplemental figure 11: Axes labels should be labeled.

We have labeled the axes for the current Fig. 3a,b.

REVIEWERS' COMMENTS:

Reviewer #1 (Remarks to the Author):

I greatly appreciate the authors' thorough response to my last review – the rewrite was extensive, but in my (presumably biased) opinion, the manuscript now reads much better and will be accessible to a larger audience. All of my remaining comments are minor.

- p2 lines 42-44 While the authors clearly know what they are talking about and what is written is correct, this is one of a few spots in the writing that could accidentally reinforce misconceptions about the limits to natural selection in small population that I have found to be common. The effect of N on beneficial effect size fixed is either complex (eg with clonal interference) or minor, with large populations $N > 1/s$ actually fixing beneficial mutations with lower probability s than smaller populations that fix them with prob $1/N$. The really important effect of N , according to Kimura's equation for the probability of fixation, is all about deleterious mutations. Here and in some other spots below, and I encourage the authors to at least consider tweaks that make the paper less likely, for a casual and non-theoretically inclined reader, to accidentally reinforce existing misconceptions that beneficial mutations are at least as important with respect to limits posed by N . I realize that the last section of the Results deals directly with this question, but think it could do with more and earlier reinforcement.

- p2 line 56. this hypothesis -> our hypothesis

- p3. I found κ confusing here. Its derivation in the Methods requires the auxiliary assumption that the DFE for beneficial mutation is exponential. Empirical data on this point is mixed, eg see distribution (Rokyta et al. 2008; Levy et al. 2015), and this questionable auxiliary assumption is not mentioned in the Results section. I think it is more intuitive to first refer directly to the beneficial mutation rates, and then perhaps mention that IF one can use an exponential to quantify the ratio of mutations rates, then one gets the expression with κ .

- lines 109-111. I would like to see more heuristic explanation here of the effect of ϵ .

- lines 116-119. It seems worth noting that the effect of κ is weaker than that of ϵ or n , appearing as it does only a log form; see my first bullet point above re the opportunity to correct or at least fail to reinforce a misconception. A similar opportunity arises re the "slight" difference lines 201-207.

- line 388 Figure 2 -> Figure 3

Signed,

Joanna Masel

Levy, SF, Blundell, JR, Venkataram, S, Petrov, DA, Fisher, DS, and Sherlock, G. 2015.

Quantitative evolutionary dynamics using high-resolution lineage tracking. *Nature* 519:181-186.

Rokyta, DR, Beisel, CJ, Joyce, P, Ferris, MT, Burch, CL, and Wichman, HA. 2008. Beneficial Fitness Effects Are Not Exponential for Two Viruses. *Journal of molecular evolution* 67:368-376.

Reviewer #2 (Remarks to the Author):

The authors have done significant work to address my comments, as well as those of other reviewers. I think the manuscript is ready for publication.

Reading their responses, one final thing stood out. They added a following sentence:

"This fitness landscape was used because the fitness landscape where logic calculations are under selection lack small-effect deleterious mutations, which would preclude the observation of drift robustness."

However, as far as I can see, this is not entirely true. The same small-effect mutations, which affect the speed of replication, would still be present if the logic was on. If anything they the effect would be (relatively, comparatively) smaller. Optimizing the genome copying would still be a small-effect mutation just like in the present setup. However, there would be many more large-effect mutations as well. I can sort of see why this would not be ideal, but I'm not fully convinced why. So, a better argument is still needed here.

Reviewer #3 (Remarks to the Author):

The presentation of the paper is much improved. I remain convinced that the science in this paper is of high interest and should be published in *Nature*.

I think that the paper could still be improved by increasing the readability of the figures. Many of the figures cannot be understood even superficially without a careful reading of the caption, and this could be substantially improved by more descriptive axes labels or sub-figure titles. Here are a few minor suggestions:

Figure 2: The figure at least implicitly has higher fitness genotypes higher on the page. But this useful metaphor falls apart with the placement of the top left (1 - s - e) oval. If that were moved downwards, the metaphor could be preserved and the reader get the right idea more easily.

Figure 3: add k value to left and n value to right as titles

Figure 4a—We can't see red and grey at all in this graph. It is impossible to tell the

difference between the large and small population results here.

Figure 4d: These box plots don't match the distribution in 4a, although from the description it seems they should. I'm not sure what is meant here.

Figure 5: Improve the y-axis label. Maybe something like "Relative fitness after 10^3 generations at $N=50$ " or "Relative fitness after drift robustness test" or similar.

Figure 7: I'm not sure what "maintained beneficial mutational effects" are. Moreover, lethal mutations are deleterious, by the definition of the words. If you want to distinguish sub-lethal deleterious mutations from lethal ones, it requires more words.

Figure 4 and line 152: Selection coefficients of 5% are fairly large, relative to the population sizes considered in this paper. Why was this value chosen as the boundary for small effect mutations?

We thank the reviewers for their additional comments. Below, we have listed the comments given by each reviewer in italics and wrote our responses below each comment in bold font. Our changes to the main text are denoted in blue.

Reviewer #1 (Remarks to the Author):

I greatly appreciate the authors' thorough response to my last review – the rewrite was extensive, but in my (presumably biased) opinion, the manuscript now reads much better and will be accessible to a larger audience. All of my remaining comments are minor.

- p2 lines 42-44 While the authors clearly know what they are talking about and what is written is correct, this is one of a few spots in the writing that could accidentally reinforce misconceptions about the limits to natural selection in small population that I have found to be common. The effect of N on beneficial effect size fixed is either complex (eg with clonal interference) or minor, with large populations $N > 1/s$ actually fixing beneficial mutations with lower probability s than smaller populations that fix them with prob $1/N$. The really important effect of N , according to Kimura's equation for the probability of fixation, is all about deleterious mutations. Here and in some other spots below, and I encourage the authors to at least consider tweaks that make the paper less likely, for a casual and non-theoretically inclined reader, to accidentally reinforce existing misconceptions that beneficial mutations are at least as important with respect to limits posed by N . I realize that the last section of the Results deals directly with this question, but think it could do with more and earlier reinforcement.

We have re-written the above section as follows:

In a large population (defined here such that its effective population size is larger than the inverse of every selection coefficient in the landscape), natural selection will ultimately lead to the fixation of all beneficial mutations. In a small population, while selection may also lead to the fixation of these beneficial mutations, weakened purifying selection inherent to small populations will result in the subsequent *loss* of these beneficial mutations. Thus, while a large population can maintain itself at the top of the fitness peak, a small population is unable to maintain fitness due to an increased rate of fixation of slightly-deleterious mutations. Therefore, this small population will not occupy the top of the fitness peak, but some lower area where the fixation of slightly-beneficial mutations and the fixation of slightly-deleterious mutations balance out (Goyal et al. 2012).

- p2 line 56. this hypothesis -> our hypothesis

Fixed.

- p3. I found kappa confusing here. Its derivation in the Methods requires the auxiliary assumption that the DFE for beneficial mutation is exponential. Empirical data on this point is mixed, eg see distribution (Rokyta et al. 2008; Levy et al. 2015), and this questionable auxiliary assumption is not mentioned in the Results section. I think it is more intuitive to

first refer directly to the beneficial mutation rates, and then perhaps mention that IF one can use an exponential to quantify the ratio of mutations rates, then one gets the expression with kappa.

We have added a mention to the Results of the assumption that beneficial mutations are exponentially-distributed in our derivation of the critical population size. We have also re-wrote the section of the Methods deriving κ to first derive the existence of a transition between the two fitness peaks without assuming an explicit distribution. Then, we introduce the assumption that beneficial mutations are exponentially-distributed and derive κ .

- lines 109-111. I would like to see more heuristic explanation here of the effect of epsilon.

We have added the following explanation to the text:

In other words, small populations will only preferentially evolve towards the drift-robust fitness peak if the trade-off between drift robustness and fitness is not too severe. If the drift-robust peak results in extremely low fitness, the small population will evolve as far up the drift-fragile peak as it can while maintaining fitness.

- lines 116-119. It seems worth noting that the effect of kappa is weaker than that of epsilon or n, appearing as it does only a log form; see my first bullet point above re the opportunity to correct or at least fail to reinforce a misconception. A similar opportunity arises re the “slight” difference lines 201-207.

We have added the following text to this section:

We should note here that κ has a weaker influence on N_{crit} than ϵ or n . This lesser influence, due to the equation containing $\log \kappa^{-1}$, exists because there is only a slight difference in the fixation probability of beneficial mutations between small and large populations. The relevant difference comes down to the lack of maintainability of these beneficial mutations in small populations, an effect captured by n , the number of beneficial mutations required to reach the drift-fragile peak.

As for the “slight” difference on lines 201-207, in attempting that analysis, we tried to only include those beneficial mutations whose fitness was maintained.

- line 388 Figure 2 -> Figure 3

Fixed.

Signed,

Joanna Masel

Levy, SF, Blundell, JR, Venkataram, S, Petrov, DA, Fisher, DS, and Sherlock, G. 2015. Quantitative evolutionary dynamics using high-resolution lineage tracking. *Nature* 519:181-186.

Rokyta, DR, Beisel, CJ, Joyce, P, Ferris, MT, Burch, CL, and Wichman, HA. 2008. Beneficial Fitness Effects Are Not Exponential for Two Viruses. *Journal of molecular evolution* 67:368-376.

Reviewer #2 (Remarks to the Author):

The authors have done significant work to address my comments, as well as those of other reviewers. I think the manuscript is ready for publication.

Reading their responses, one final thing stood out. They added a following sentence:

“This fitness landscape was used because the fitness landscape where logic calculations are under selection lack small-effect deleterious mutations, which would preclude the observation of drift robustness.”

However, as far as I can see, this is not entirely true. The same small-effect mutations, which affect the speed of replication, would still be present if the logic was on. If anything they the effect would be (relatively, comparatively) smaller. Optimizing the genome copying would still be a small-effect mutation just like in the present setup. However, there would be many more large-effect mutations as well. I can sort of see why this would not be ideal, but I’m not fully convinced why. So, a better argument is still needed here.

While it is true that genome copying can still be optimized in the traditional logic-9 Avida fitness landscape, the presence of the logic functions really do alter the fitness landscape such that small-effect deleterious mutations are lacking. In a recent experiment on the logic-9 fitness landscape for another project, we observed that our genotypes’ likelihood of small-effect deleterious mutations was less than 15%. By the standards of this manuscript, that would make these genotypes automatically drift robust, although they did not evolve at a small population size.

This lack of small-effect deleterious mutations occurs because there is strong antagonistic pleiotropy and tradeoffs between optimizing genome copying and evolving complex traits in Avida. If a given locus in the genome evolves to contribute to a complex trait, it does not then contribute to genome copying. Thus, the percentage of mutations that alter logic functions increases and the percentage of mutations that alter genome copying decreases. We have added the following text to the above sentence explaining this:

This fitness landscape was used because the fitness landscape where logic cal-

culations are under selection lack small-effect deleterious mutations, which would preclude the observation of drift robustness. This lack of small-effect deleterious mutations occurs due to antagonistic pleiotropy and trade-offs between the logic functions and genome replication. Because there are a fixed number of loci in the genome, the more loci dedicated to the logic functions, the fewer loci dedicated to genome replication. Therefore, because most loci are dedicated to logic functions, and mutations to these loci are strongly-deleterious, there are few small-effect deleterious mutations in the logic-function fitness landscape.

Reviewer #3 (Remarks to the Author):

The presentation of the paper is much improved. I remain convinced that the science in this paper is of high interest and should be published in Nature.

I think that the paper could still be improved by increasing the readability of the figures. Many of the figures cannot be understood even superficially without a careful reading of the caption, and this could be substantially improved by more descriptive axes labels or sub-figure titles. Here are a few minor suggestions:

Figure 2: The figure at least implicitly has higher fitness genotypes higher on the page. But this useful metaphor falls apart with the placement of the top left (1 - s - e) oval. If that were moved downwards, the metaphor could be preserved and the reader get the right idea more easily.

Fixed.

Figure 3: add k value to left and n value to right as titles

We have made the above change.

Figure 4a—We can't see red and grey at all in this graph. It is impossible to tell the difference between the large and small population results here.

This figure for the overall distribution of fitness effects was included to show that both small-population and large-population avidian DFEs have roughly the same shape as biological DFEs. Any informative differences between the two can be detected in 4b. We could show a figure of a portion of the DFE, but this then loses the value of giving readers an idea of the overall shape of avidian DFEs.

Figure 4d: These box plots don't match the distribution in 4a, although from the description it seems they should. I'm not sure what is meant here.

We agree that our description here is unclear. These boxplots show the *mean* relative fitness of each mutation for each of the 100 small-population and 100 large-population genotypes (i.e., a mean relative fitness is calculated for each genotype and the boxplots show the median of these means). So while these

data do come from the DFE shown in 4a, the connection is not clear. We have corrected the caption and y-axis label to state it is the mean relative fitness.

Figure 5: Improve the y-axis label. Maybe something like “Relative fitness after $10\hat{3}$ generations at $N = 50$ ” or “Relative fitness after drift robustness test” or similar.

We have changed the relevant axis labels in both Fig. 5 and Fig. 8 to “Relative Fitness after Drift Test”.

Figure 7: I’m not sure what “maintained beneficial mutational effects” are. Moreover, lethal mutations are deleterious, by the definition of the words. If you want to distinguish sub-lethal deleterious mutations from lethal ones, it requires more words.

We have added the following line to Figure 7a’s caption: “See main text and methods for our definition of maintained beneficial mutations.” We have also clarified our description of lethal vs. deleterious mutations in Figures 4 and 7 and in the text by specifying that deleterious mutations includes only viable mutations.

Figure 4 and line 152: Selection coefficients of 5% are fairly large, relative to the population sizes considered in this paper. Why was this value chosen as the boundary for small effect mutations?

We chose this boundary, and not, say, $\frac{1}{N} = \frac{1}{100} = 1\%$ because these genotypes often accumulate multiple deleterious mutations during their drift-induced fitness declines. Additionally, their genotypes’ high genomic mutation rate implies they have an effective population size lower than 100, so a boundary percentage greater than 1% should be used, but it is not clear which boundary should have been used. In a previous iteration of this work, a boundary of 2% seemed to explain a genotype’s fitness decline almost as well as 5%. However, to avoid issues with statistics and testing many boundaries, we stuck with just one boundary for these experiments.